# Characterizing some major Archean faults at depth in the Superior craton, North America

David B. Snyder[1,2], Jack M. Simmons[2], John A. Ayer[1,2], Mostafa Naghizadeh[2], Ademola Q. Adetunji[2], Taus R. C. Jørgensen[2], Graham J. Hill[3], Eric A. Roots[2], and Saeid Cheraghi[2]

[1] Harquail School of Earth Sciences, Laurentian University, 935 Ramsey Lake Road, Sudbury, ON, Canada, P3E 2C6

[2] Metal Earth, Mineral Exploration Research Centre (MERC), Laurentian University, 935 Ramsey Lake Road, Sudbury, ON, Canada, P3E 2C6

[3] Institute of Geophysics, Czech Academy of Sciences, Prague, Czech Republic

*Correspondence to*: David B. Snyder (dbsnyder1867@gmail.com)

**Abstract.** The geometry of ancient (2.75–2.65 Ga) faults at depth can only be mapped in detail by high-resolution geophysical surveys such as seismic reflection profiling. Recent deep (35–48 km) reflection profiles acquired throughout the Archean southern Superior craton of North America provided such data with which to map in 3-D some major shear zones, many of which are associated with significant orogenic gold or volcanogenic massive sulfide (VMS) deposits. Most faults are (re)interpreted as low-angle (<35°) thrusts; a few appear as sub-vertically (>75°) aligned truncations of prominent reflectors. Asymmetry of reflectors suggests that the sub-vertical faults may have originated as 2.75–2.70 Ga syn-volcanic leaky transform faults. We relate thrust structures primarily to the dominant phase of folding and horizontal shortening strain that occurred at 2.72–2.66 Ga during regional crustal deformation, mineralization, and peak metamorphism, associated with terrane accretion. Palinspastic restoration near Timmins, Ontario, indicates 40 km of horizontal shortening. Previous mapping indicates that deformation after this orogenic shortening event resulted in modest lateral movement. Coincident magnetotelluric (MT) surveys indicate pervasive conductive minerals, such as graphite/carbon and sulfide, exist within the mid-crust and in near-vertical channels within the more brittle and resistive upper crust. Many such channels, but not all, coincide with fault zones and mineral deposits. Palinspastic and paleomagnetic-based reconstructions suggest many faults had multiple periods of activity with evolving vertical to horizontal offsets. Some prominent faults appear paired, partitioning normal and oblique strains on vertical shear zones and dipping thrust zones, respectively.

## 1. 1 Introduction

Major Archean-age faults typically require more cryptic description than do neotectonic faults due to an accumulation of overprinting structures, poor surface exposures, extensive metamorphism, or a lack of supporting syn-tectonic observations such as earthquakes. Conversely, observations of these faults at depth become enhanced by the stronger propagation of

seismic waves within their host crystalline rocks. High-resolution geophysical techniques, including reflection profiling,

have consequently been employed to resolve the geometry of ancient crustal-scale faults, particularly within the Superior craton of North America (e.g., Calvert et al., 2004; Snyder et al., 2008), where shear zones and faults are spatially associated with world-class gold deposits (e.g., Cadillac-Larder Lake and Porcupine-Destor faults; Poulsen, 2017; Dubé and Mercier-Langevin, 2020).

The crust of the Superior craton (Fig. 1), the world's largest exposure of Archean crust, has historically been recognized as

consisting of three layers, colloquially known as the upper-crustal "greenstone" (granite-greenstone) layer, the middle crustal "TTG" (tonalite-granodiorite-trondhjemite ) layer, and the lower crustal layer (Percival and West, 1994; Bellefleur et al., 1995; Calvert and Ludden, 1999; Benn, 2006; Benn and Peschler 2005; Percival et al., 2012).

Seismologists similarly consider the crystalline crust as comprising three layers when analyzed using refraction methods or alternatively, one layer when using receiver functions (Rudnick and Fountain, 1995; Mooney, 2010; Szwillus et al. 2019).

Adopting either a two or three layer model results in good matches between modeled Vp velocities and estimated bulk weight percent $SiO_2$ content (Hacker, Kelemen, and Behn, 2015; Artemieva and Shulgin, 2019). Globally averaged velocities for each layer are Vp = 5.6–6.4 km/s (67 % $SiO_2$) for the upper crust, Vp = 6.5–6.9 km/s (64 % $SiO_2$) for the middle crust, Vp = 7.0–7.4 km/s (53% $SiO_2$) for the lower crust, and Vp = 7.8–8.2 km/s for the uppermost mantle (Mooney, 2010; Hacker, Kelemen, and Behn, 2015).

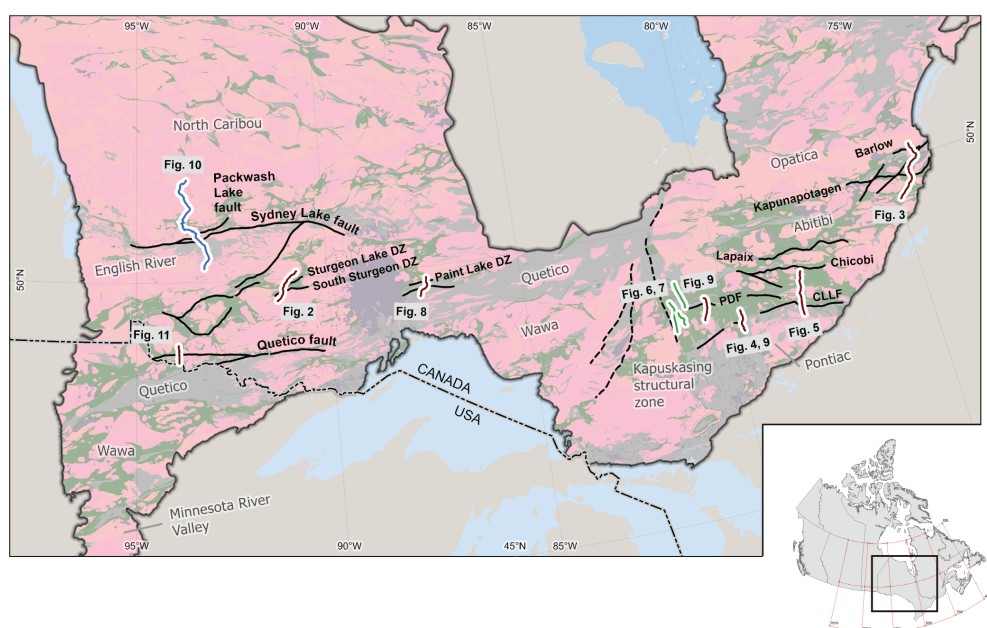

**Figure 1. Simplified basement geology map of the southern Superior craton (modified from Percival et al., 2006). Green indicates dominantly mafic greenstone belts, pink granitic plutons and gneiss, and gray metasedimentary rocks. Solid black lines are major faults. Major terranes are labelled but no boundaries are shown as these are disputed and evolving. Locations of seismic reflection profiles used in this study are shown with the corresponding figure number. CLL is the Cadillac– Larder Lake fault, PDF is the**

**Porcupine–Destor fault. Black dashed lines mark component faults of the Kapuskasing structural zone.**

Similar velocity values are modelled for the granite-greenstone terranes of the southern Superior craton (Grandjean et al., 1995; Calvert and Ludden, 1999), calibrated by an exposed crustal section at the Kapuskasing uplift (Percival and West, 1994). Variable velocities of 5.6–6.4 km/s characterize the upper crust (0–12 km). a relatively uniform middle crust (~12–30 km) has velocities of 6.4 to 6.6 km/s. A velocity increase of 0.3 km/s occurs at 30–40 km depths, across the middle crust–lower crust boundary. Lower crustal velocities increase from 6.9 km/s to 7.3 km/s at the Moho. Uppermost mantle has velocity of 8.15 km/s.

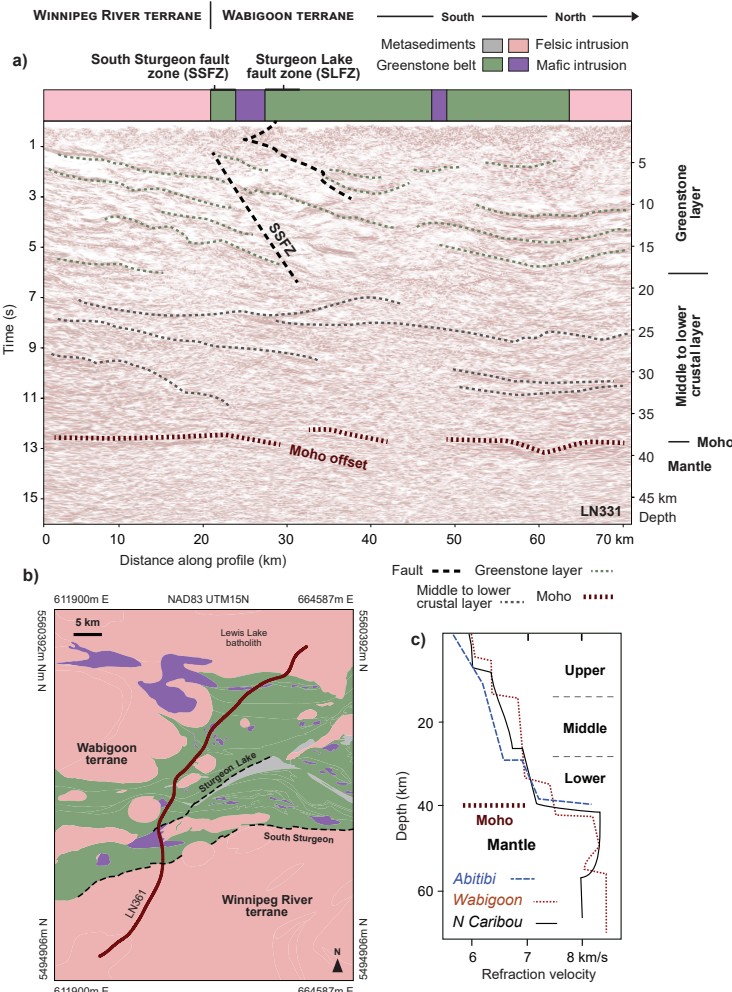

**Figure 2. Deep seismic reflection section across the terrane boundary between the Wabigoon and older Winnipeg River terranes: the Metal Earth Sturgeon transect (Fig. 1) (Simmons et al., 2024). (a) Strip geology and interpreted seismic reflection section; (b) geological map; (c) seismic wave speed versus depth from near coincident model of Mussachio et al. (2004; Grandjean et al., 1995). The depth scale in (a) assumes a constant velocity of 6 km/s to convert from two-way travel time. Details of seismic data acquisition and processing (pre-stack migration, curvelet dip coherency enhanced) described in Nazghizadeh et al., 2019). This section illustrates both broad folding and thrusting within the upper crust and layered, gently north-dipping reflectors in the middle crustal layer. Moho at 39-40 km is inferred from crossing seismic refraction models (Musacchio et al., 2004). The South Sturgeon fault zone is the interpreted terrane boundary (Ma et al., 2021). Greenstone assemblages range from 2.78 to 2.70 Ga.**

Here we assemble new and older seismic observations (with MT data: Simmons et al., 2024), consistent with previous interpretations of the layered structure of the crust, to constrain the architecture and tectonic history of ancient major faults that cut the crust, and in places the lithosphere, of the Superior craton of central North America.

The goal is to provide a consistent interpretation of recent deep seismic reflection profiles across the southern Superior craton. To that end, selected seismic sections and coincident conductivity sections taken from the Metal Earth Atlas (Simmons et al., 2024) have retained the consistent display parameters and scales. All underlying data were acquired and processed consistently except for the Red Lake and Timmins/Crawchest sections which were acquired in the early 2000's (Calvert et al., 2004; Snyder et al., 2008).

## 2 Geologic Setting

The Superior Province is the largest coherent area of Archean cratonic rocks on Earth. It consists of Eo- to Neoarchean granite-greenstone, metasedimentary, or gneissic domains that mainly assembled during the Neoarchean (e.g., Percival and Williams, 1989; Williams et al., 1992; Stott et al., 2010; Percival et al., 2012). Subprovinces have been defined based on shared structural, magmatic, geochemical, and detrital zircon characteristics (Percival et al., 2006, 2012; Stott et al., 2010), interpreted as either terranes or superterranes (Percival et al., 2012). The southern Superior Province underwent Neoarchean assembly starting ~2.720 Ga, marked by a southward-moving collisional front that first joined the Winnipeg River and Marmion subprovinces with the North Caribou subprovince (Corfu et al., 1995; Percival et al., 2012). This tectonic system featured south-verging structures, syn-deformational tonalite plutons dated to ~2.710 Ga, and sedimentation in syn-deformational basins between 2.715–2.705 Ga, such as those in the English River subprovince (Sleep, 1992; Corfu and Stone, 1998; Percival et al., 2012). Diachronous regional deformation along the southern edge of the North Caribou subprovince generally occurred between 2.720–2.705 Ga and extended into the Western Wabigoon subprovince (Corfu and Stott, 1993; Percival et al., 2006). By ~2.700 Ga, these areas experienced retrograde metamorphism and emplacement of post-deformational plutons (Sleep, 1992; Corfu and Stone, 1998). Contractional deformation extended into the Abitibi and Wawa subprovinces by~2.695 Ga, following their collision with the amalgamated northern subprovinces (Corfu and Stott, 1998; Bateman et al., 2008).

The upper crustal layer (0–12 km) of the greenstone dominated terranes of the Superior craton, which have calibrated velocities of 5.6–6.4 km/s, consists predominantly of ultramafic-felsic volcanic rocks, plutons emplaced between 2.99 and 2.70 Ga and metasedimentary deposits (Benn and Peschler, 2005; Percival et al., 2006). The preserved volcanic deposits in these terranes record multiple volcanic episodes (assemblages), each characterized by submarine mafic lavas and minor volcaniclastic deposits (± other ultramafic and intermediate to felsic lithologies; Ayer et al., 2002). Pre-2.8 Ga supracrustal assemblages and gneissic rocks are widespread across the Superior and considered to be basement unconformably underlying >2.8 Ga Neoarchean assemblages. Mesoarchean supracustal assemblages are more prevalent in the northern Superior Subprovinces such as the North Caribou and Uchi, but also locally occur within the Wabigoon and Wawa (Percival

et al., 2012). The volcanic "greenstone" rocks erupted from 3.00 to 2.69 Ga across most of the Superior craton. The west-central Abitibi terrane has a particularly well-established stratigraphy, with six Neoarchean metavolcanic assemblages: the 2.750–2.735 Ga Pacaud, 2.734–2.724 Ga Deloro, 2.723–2.720 Ga Stoughton-Roquemaure, 2.720–2.710 Ga Kidd-Munro, 2.710–2.704 Ga Tisdale, and the 2.704–2.695 Ga Blake River assemblages (e.g., Ayer et al., 2002, 2005; Monecke et al., 2017). Their basal contact with 2.87–2.70 Ga tonalitic gneisses lies at 10–12 km depth (Snyder et al., 2008).

Eruption of the metavolcanic assemblages occurred coeval with emplacement of syn-volcanic plutons of gabbro and tonalite-granodiorite-trondhjemite (TTG) that form the upper part of the middle crustal layer (Matthieu et al., 2020). The transition from the upper crustal greenstone layer into the middle crustal layer corresponds to an increase in crustal velocities to 6.4–6.6 km/s (Grandjean et al., 1995; Calvert and Ludden, 1999). Anorthosite, tonalite, mafic gneiss and paragneiss form layers 0.1 to 10-km thick in the lower part of the middle layer (Percival and West, 1994). The third layer, the lowermost crust, is not exposed but hypothesized to consist of dense, mafic rocks with pyroxene- and amphibole-rich minerals (White et al., 2003; Benn and Peschler, 2005).

Greenstone belts separate plutons and older gneiss domains (Fig. 1). These upper crustal plutonic rocks have generally been classified as syn-volcanic (>2.70 Ga), syn-tectonic (2.72–2.66 Ga), or late intrusions, although some intrusive complexes formed over 90 million years by multiple intrusive events (Ayer et al., 2005; Beakhouse, 2011; Matthieu et al., 2020; 2024). Isotopic zircon dating indicates that syn-volcanic plutons crystallized coeval with the mafic volcanic rocks whereas syn-tectonic plutons typically core folds or crystallized synchronous with formation of sedimentary basins, such as those hosting the English River, Porcupine or Quetico metasediments (e.g., Žák et al., 2023). Late intrusions (post 2.7 Ga) have more hydrous geochemistry and form sanukitoid batholiths and small syenite or pyroxenite bodies (Mole et al., 2021).

Gneissic domains that core terranes of the southern Superior craton include rocks of diverse ages that include the 3.05–2.72 Ga North Caribou/Opinaca, the 3.32–2.66 Ga Winnipeg River (Opatica), the 3.01–2.66 Ga Wabigoon/Marmion, the 2.79–2.69 Ga Wawa/Abitibi, and the 3.5–2.65 Ga Minnesota River Valley terranes (Bjorkman et al., 2024; Snyder and Thurston, 2024). These distinct terranes may have accreted successively to the southern margin of the Superior craton (Percival et al., 2012), but Mole et al. (2021) indicate more complexity. Compiled geochronology suggests different pre-2.9 Ga histories for the Opatica, Wawa, Wabigoon, and Winnipeg River terranes (Bjorkman et al., 2024). The Wabigoon terrane has a broadly similar range of Neoarchean ages (2.790-2.700 Ga) but differs in localized evidence of Mesoarchean (2.8-3.0 Ga) inheritance in isotopic signatures indicating at least localized presence of an older crustal substrate this is also evident in the Winnipeg River terrane which surrounds the Wabigoon to the north and east. Further to the north the Uchi terrane consists of Meso- to Neoarchean supracrustal successions (3.0-2.7 Ga), in tectonic juxtaposition with the 2.7 Ga English River metasedimentary terrane to the south.

Dipping seismic reflectors observed on Lithoprobe and more recent deep seismic reflection sections (Fig. 3), which have been interpreted in several locations as lower crust underthrust in a general north direction, permit diverse implied tectonic histories (Calvert et al., 1995; White et al., 2003; Matthieu et al., 2020). This underthrusting thus has been related to serial subduction zones (Percival et al., 2006), subcretion to the leading edge of a drifting proto-craton (Bédard and Harris, 2014)

or progressive rifting (Bjorkman et al., 2024; Mints, 2017; Mole et al., 2021). Many of these interpretations agree that the oldest (North Caribou) block began drifting south and east at ca. 2.72 Ga so that dipping crustal structures record tectonic
imbrication ahead of and beneath the drifting proto continent (Snyder and Thurston, 2024). One serial subduction model for the western part of the Superior craton includes at least four subduction zones: North Superioran, Uchian, Shebandowanian, and Minnesotan (Card, 1990).

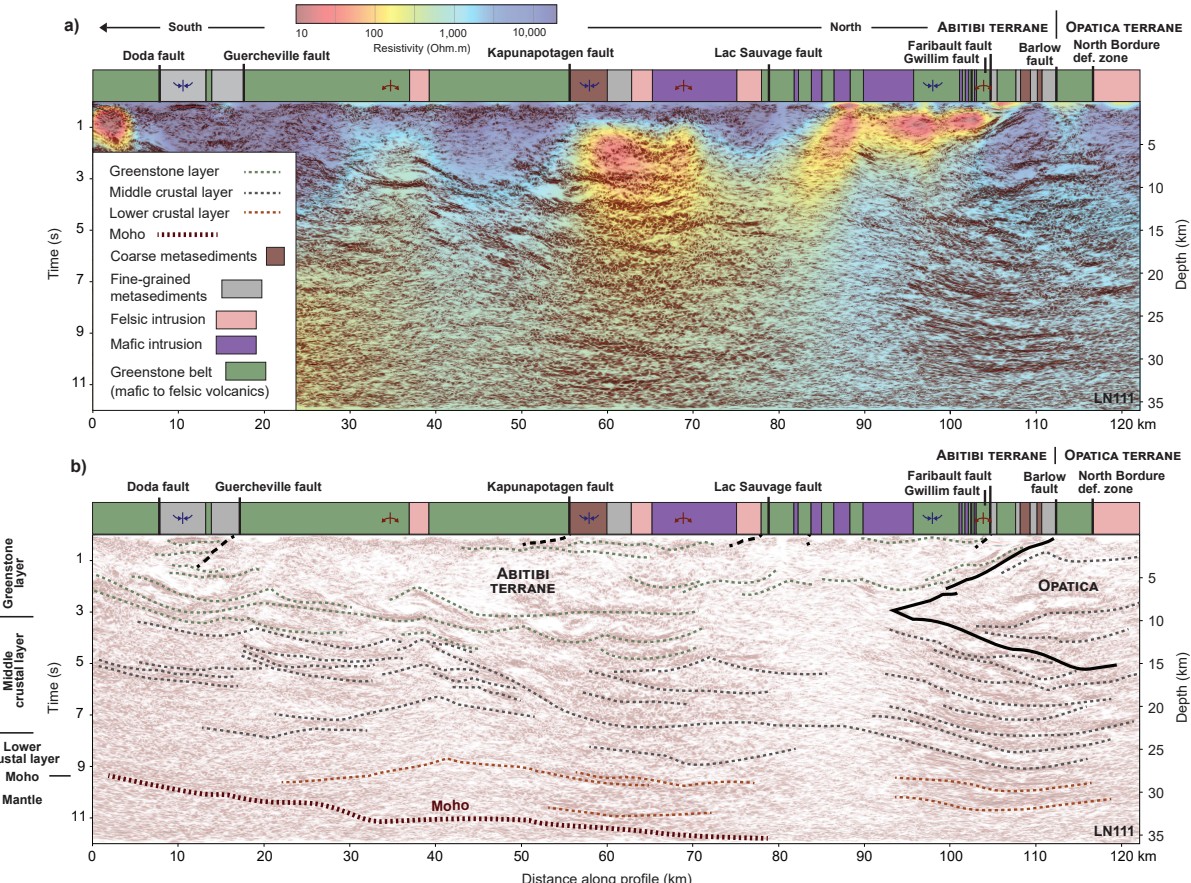

**Figure 3. Deep seismic reflection section across the terrane boundary between the Abitibi and older Opatica terranes: the Metal Earth Chibougamau transect (Fig. 1). (a) Strip geology, seismic reflection section and conductivity model section (Simmons et al., 2024); and (b) interpreted seismic reflection section. Green, black, red dotted form lines mark upper, middle, lower crustal reflectors, respectively. Heavy dotted line marks interpreted base of Abitibi crustal rocks (Moho). Note step in Moho (red bars) from 38 to 34 km as determined by 3D Ps receiver function analysis (station CHGQ in Snyder et al., 2022). This section illustrates**
**both early (D1), broad folding in the upper crust and late-stage (D3) collisional structures in the lower crust underthrusting of juvenile Abitibi crust and wedging of more evolved Opatica (North Caribou) crust between the more juvenile lower crust.**

Mapping has demonstrated that three to five phases of deformation occurred across the southern Superior craton (Percival et al., 2006). The cryptic, earliest (D1) phase involves horizontal shortening that produced gentle fold interference and

sufficient topography to produce widespread sedimentary basins such as those represented by the English River, Quetico and Pontiac sediments and coeval metasedimentary assemblages within the volcanic subprovinces (e.g., the Porcupine assemblage within the Abitibi subprovince). Fold interference patterns imply a second, dextral phase followed relaxation extension (e.g., Davis et al., 1988; Calvert et al. 2004; Bleeker, 2015; Žák et al., 2021; Tóth et al., 2023, 2025). The most prominent deformation, the Kenoran or D3 phase of north-south shortening at 2.72-2.66 Ga, generally coincided with peak metamorphism and gold mineralization or remobilization (Dubé and Mercier-Langevin, 2020). Later, more localized deformation phases include dextral and sinistral transpressive phases with modest displacements that may represent conjugate shearing; this deformation has associated distinct (Timiskaming-type) polymict conglomerate and clastic deposits (e.g., Bleeker, 2015; Bedeaux et al., 2017; Žák et al., 2021).

The post-2.6 Ga stabilized Superior craton records little subsequent internal deformation except for the Kapuskasing uplift, a 500 km long, east-verging thrust fault-bounded block (Fig. 1) (Percival and West, 1994). Cooling patterns observed within the uppermost 20 km of granulite-facies rocks suggest uplift and rotation of the hanging wall (western) block occurred at about 1.9 Ga (Percival et al., 2012). Paleoproterozoic hydrothermal activity occurred one hundred kilometers to the east, where evidence for quartz veining and possible upgrading of Au deposits suggests incursion of mantle-derived melts at 1.88 and 1.75–1.70 Ga that form part of the Circum-Superior Large Igneous Province (Zhang et al., 2014; Herzberg, 2022).

## 3 Faults observed on seismic sections: reflections

Determining the geometry of faults at depth requires high-resolution geophysical methods such as seismic reflection profiling. Interpreters generally infer faults from aligned breaks in prominent reflectors, rarely from a continuous reflector itself. Several new data acquisitions relevant to the mapping of faults and related crustal structures have occurred in the southern Superior craton since synthesis of the Lithoprobe program (e.g., Percival et al., 2006; 2012). Crustal-scale and some high-resolution seismic reflection profiling now covers much of the Abitibi and Wabigoon terranes (Simmons et al., 2024). This newer profiling can resolve structures as small as several hundred meters and a kilometer in the upper and lower crust, respectively (Snyder et al., 2008; Nazghizadeh et al., 2019; Haugaard et al., 2021). The Moho and uppermost mantle structures occur deeper than some of these sections, but acquisition of those sections was designed for future reprocessing using extended correlation should deeper images be of particular interest (e.g., Okaya and Jarchow, 1989). New coincident broadband magnetotelluric (MT) soundings along transects augment the mostly long-period MT (LMT) soundings acquired by Lithoprobe (Jones et al., 2014; Hill et al., 2021) and provide independent characterization of crustal structures. Three-dimensional analysis and modelling at lithospheric scale of all these data types provides a common reference frame (Snyder et al., 2021; Snyder and Thurston, 2024).

Seismic reflection sections collapse information from a 3-D volume onto a 2-D plane. All crustal reflectors with appropriate strike and dip lying within a triangular prism with an apex axis coincident with the survey line and a base several kilometers wide will appear on the section. Thus, sets of reflectors that appear to cross may lie on opposite sides of the survey line.

Careful cross dip analysis and 3-D migration can restore correct geometries but requires considerable processing effort (e.g., Calvert et al., 2004, Cheraghi et al., 2022; Fam et al., 2023). Where available, such sections aided our fault interpretation. Zones of apparently reduced reflectivity may thus result from either reflectors that occupy only part of the subsurface volume being imaged or changes in rock properties that make rock layers more homogenous or both.

Our interpretation of the seismic sections (see Fig. 2 or 3 for example) uses form lines to emphasize relatively large amplitude, laterally continuous (minimum several kilometers) reflectors. Green form lines indicate upper crustal reflectors in the presumed brittlely deforming and generally resistive (Hill et al., 2021) mafic volcanic and plutonic rocks of the various Superior greenstone belts crossed by transects. Black form lines indicate reflectors within the presumed plastically deforming and conductive gneiss of TTG composition in the mid-crust; in the Kapuskasing uplift, such reflectivity appears in syn-volcanic and younger plutonic rocks, which were strongly reworked at ca. 2.660–2.610 Ga into sub-horizontal gneissosity (Percival and West, 1994). Red form lines indicate presumably strong mafic-ultramafic lowermost crust composed of pyroxenites, amphibolites or garnet granulites. Thick dashed lines map breaks in reflectors and hence the interpreted continuation of named, mapped surface faults at depth. We further assume that the upper crustal 'greenstone' layer appears generally more resistive in coincident conductivity models, whereas TTG gneisses of the mid-crust appear moderately conductive, and faults are variably conductive (Hill et al., 2021, 2025; Roots et al., 2024).

As noted previously, petrogenesis of plutons in the southern Superior craton has historically been distinguished by age as: (1) synvolcanic with the greenstone mafic lava assemblages (2.790–2.685 Ga), (2) syntectonic with the major deformation and metamorphism phase (2.685–2.667 Ga), and (3) late, during final stages of collision and stabilization (circa 2.5 Ga cratonization) of the crust and uppermost mantle (Beakhouse 2011; Ayer et al., 2002; Matthieu et al., 2020, 2024). Tectonic evolution of major crustal faults will be discussed using the same three-part temporal framework. Particular seismic/MT transects provide the best examples and insights into each crustal layer and thus the characteristics of faults deforming the crust in each layer.

### 3.1 Syn-volcanic faults

Fault-related structures observed today within the 'granite-greenstone' upper crustal rocks generally form either steep (>75°) zones of truncated or disrupted reflectors or low-angle (<35°) thrust faults. Mafic–felsic, juvenile, polycyclic volcanic rocks deposited between 2.99 and 2.70 Ga typically form broad upright folds in greenschist metamorphic facies (Ayer et al., 2002, 2005; Benn and Peschler, 2005). The low-angle thrusting, stacking and broad folding of these volcanic units clearly post-dates their emplacement. In contrast, the steep (>75°) zones of disrupted reflectors may also indicate intermittent faulting synchronous with eruption of the volcanic units. When considering deeper continuity of the fault zones within the middle crust, the original contact between folded resistive greenstone belt mafic volcanic rocks in the upper crust and the more conductive banded gneissic middle crust becomes important. This seldom-observed contact remains controversial. It forms a tectonic boundary, a detachment, if the greenstone belt represents a tectonically emplaced nappe complex (e.g., Dimroth et al., 1985; Calvert et al., 1995), or a stratigraphic boundary if the volcanic rocks erupted directly onto plutons or gneiss

complexes (e.g., Goodwin and Smith, 1980; Thurston, 2002; Benn and Peschler, 2005). Thrust faults and wedges characterize the former; through-going steep faults the latter.

    An example of a steep deformation zone appears most convincingly where the Cadillac-Larder Lake (CLL) fault zone cuts the 2.704–2.695 Ga Blake River assemblage near Larder Lake, Ontario (Fig. 4). Bedeaux et al., (2017) described a structural segmentation of the CLL fault zone east of Larder Lake, with each straight segment having a distinct orientation, foliation

and lineations. The earliest lineations are steep, atypical of strike-slip faults. They argued that this segmentation predated (>2.680 Ga) sediments in which the earliest structures were measured. Later structures indicate a more uniform regional strain field. No geochronological constraints are currently known from this pre-sediment, syn-volcanic period, thus multiple interpretations of CLL fault zone origins are possible.

    Globally the most recognized syn-volcanic faults occur as 'growth faults' in volcanoclastic sedimentary basins (e.g., Childs

et al., 2003). Characteristics include truncation of basinal sedimentary layers (reflectors) where such faults bound the basin or sharp changes in layer thickness across the growth faults. Both result from either basin subsidence or down-dropping of a basinal block during active sedimentation, but strike-slip displacements can produce similar effects. Observations of lineations on fault surfaces can resolve this ambiguity if available. Layers of volcanic lava flows and airfall deposits can form identical structures, but plutonism typically accompanies such volcanism so sill-like intrusions can inflate older, deeper

(mid-crustal) layers. If such intrusion is asymmetrical, layered reflectors may mimic basinal growth faults.

    On the Larder Lake seismic section, individual reflectors or sequences do not align across the disrupted zone, but also do not indicate consistent vertical offsets across the zone typical of normal or reverse faults (Fig. 4). Many of the reflectors are truncated at the deformation zone. This reflector asymmetry, combined with the observed CLL fault segmentation, suggests largely pull-apart displacements typical of transfer faults, leaky transform faults or incipient rifts in modern oceanic

lithosphere (e.g., Baxter et al., 2020; Lorin Fassbender et al., 2024). The observed asymmetry of reflector density across the CLL suggests that greater deposition of volcanic flows and more intrusion of sills occurred north of the fault zone during a weakly extensional strain regime.

    The near-vertical fault zones cut most reflectors in both the resistive, upper crustal 'granite-greenstone' layer and the more conductive, horizontally layered mid-crustal 'TTG' layer (Fig. 4). Eruption of the volcanic assemblages probably occurred in

association with emplacement of syn-volcanic plutons of gabbro and tonalite-trondhjemite-granodiorite (Ayer et al., 2002). Many of these near-vertical fault zones may thus have originated coeval with the volcanism and be very long-lived (>50 million years), with multiple reactivations in diverse stress regimes. The Peggy Ridge fault in the Lau Basin in the southwest Pacific Ocean may represent a modern analogue of such faults (Baxter et al., 2020). Within the Superior craton, many of these steep deformation zones have relatively high conductivity that varies along strike and thus represents a strongly 3-D

conductivity structure.

    We see no similar evidence for syn-volcanic faults on our other seismic sections, but asymmetrical distribution of the Kidd-Munro and Blake River assemblages across the dipping west-central Porcupine-Destor fault may mark a possible candidate (Figs. 6 and 7) (Ayer et al., 2002). Such asymmetry can also be caused by differential erosion or preservation.

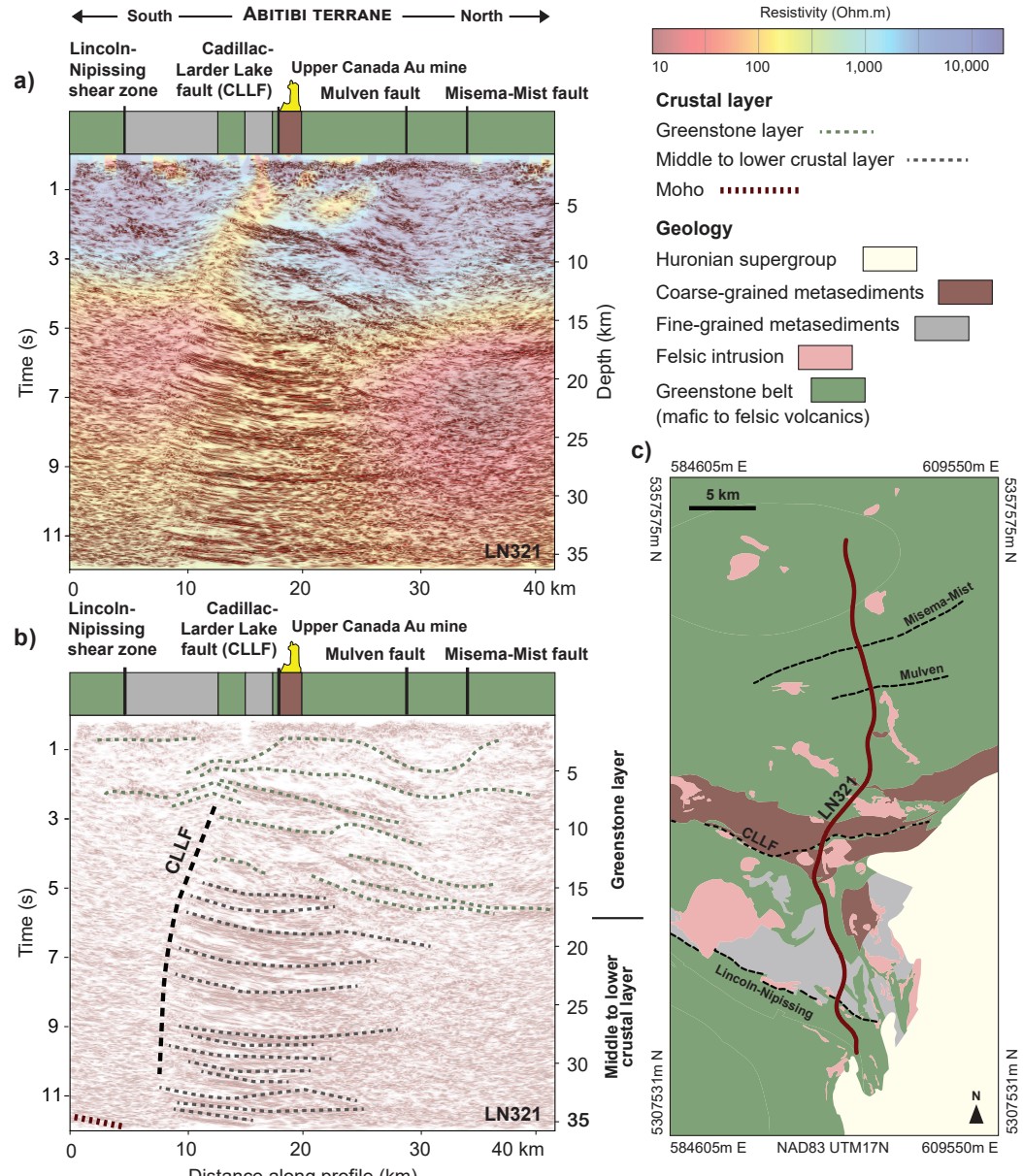

**Figure 4. Deep seismic reflection section across the Cadillac–Larder Lake fault zone in the central Abitibi terrane: the Metal Earth Larder Lake transect (Fig. 1). (a) Strip geology, seismic reflection section and conductivity model section (Simmons et al., 2024); (b) interpreted seismic reflection section; (c) Geologic location map. This section illustrates strongly layered reflectors (dotted black form lines) throughout much of the (middle) crustal layer. Moho at 38-40 km is estimated from a coincident Ps receiver function transect (Nazghizadeh et al., 2022). Interpretation of some shallow (< 2 km) reflectors benefited from cross-dip analysis (Fam et al., 2023). See Roots et al. (2022), for a more regional, alternative interpretation using older, lower-resolution seismic data.**

## 3.2 Syn-tectonic faults

Late Neoarchean metasedimentary belts within the Superior craton (e.g., English River, Quetico, Porcupine and Pontiac greywackes) record erosion during early (D1) shortening, extensional collapse about 2.75 Ga (Calvert et al., 2004) and renewed (D3) shortening culminating in peak metamorphism at 2.69–2.67 Ga (Mueller and Donaldson 1992; Leclerc et al. 2012; Bleeker, 2015; Matthieu et al., 2020).

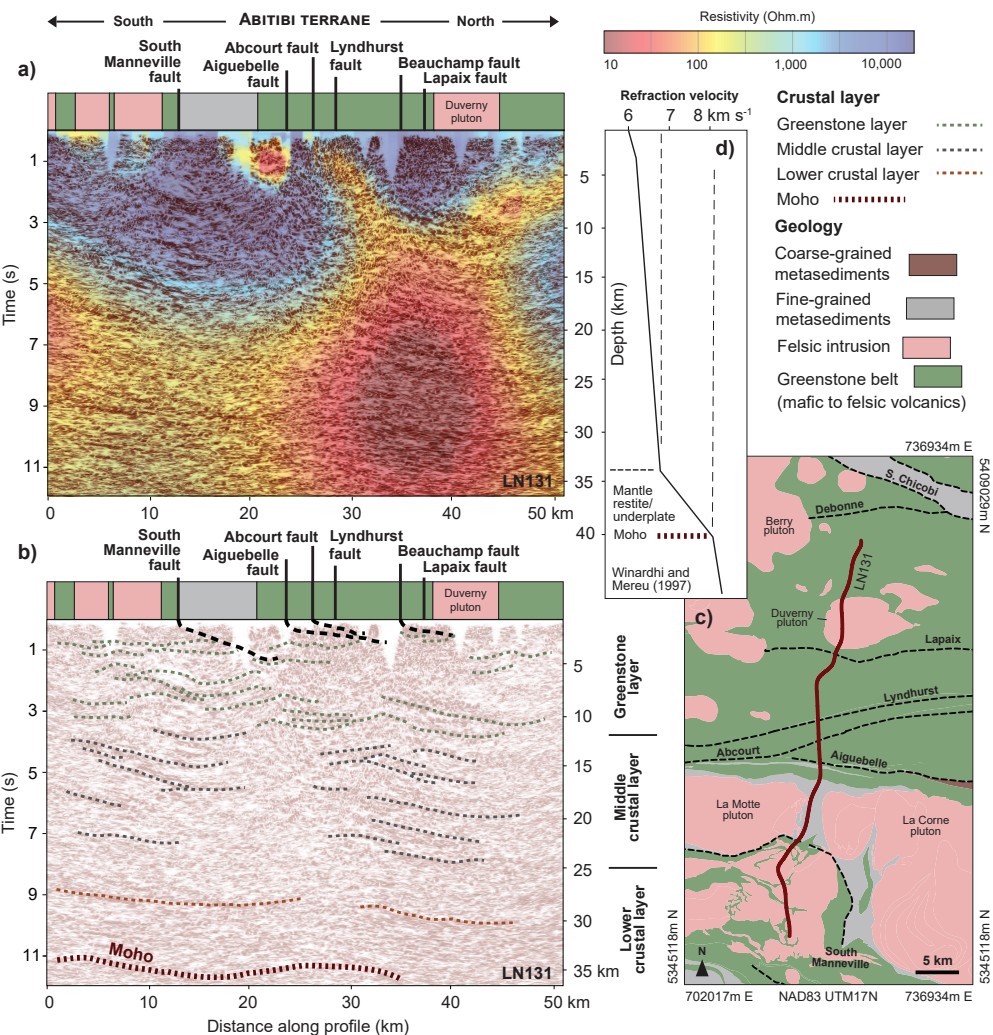

**Figure 5. Deep seismic reflection section across the central Abitibi terrane: the northern part of the Metal Earth Malartic transect (Fig. 1). (a) Strip geology, seismic reflection section and conductivity model section (Simmons et al., 2024); (b) interpreted seismic reflection section; (c) Geologic location map; (d) Velocity-depth profile (Calvert and Ludden, 1999). Seismic data processed as inFig.2. This section illustrates both broad folding and thrusting within the upper crust and strongly layered reflectors in the middle crustal layer. Moho at 38-40 km is inferred from seismic refraction models 50 km to the north (Grandjean et al., 1995). Greenstone assemblages range from 2.75 to 2.70 Ga.**

Low-angle faults observed within greenstone belts across the Superior craton fold and stack the preserved metavolcanic and metasedimentary assemblages indicating that they were active after eruption of the volcanic rocks (Figs. 5–9). Folded and thrust-stacked greenstone belts across the southern Superior craton occupy the upper crust to 10–20 km depth, based on their consistently high resistive character and an increase in parallel reflector density beneath the inferred basal intrusive or fault contact between the greenstone assemblages and 2.87–2.70 Ga tonalitic gneisses of the middle crust (e.g., Snyder et al., 2008).

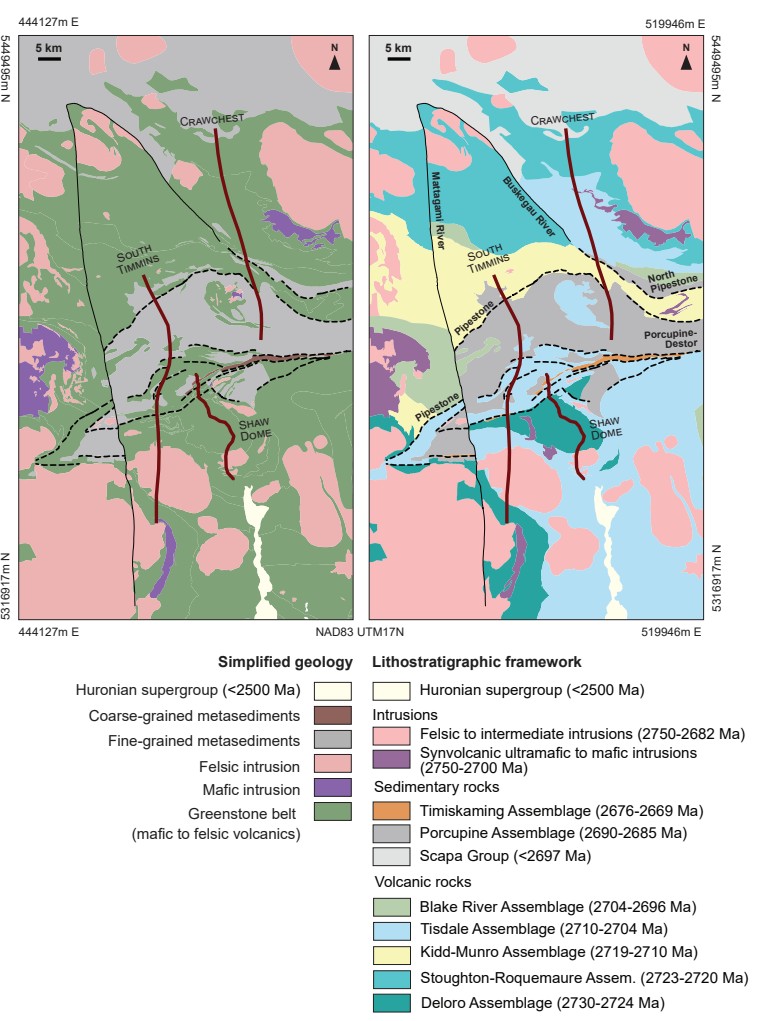

**Figure 6: Geological and assemblage (lithostratigraphic) maps for Crawchest and South Timmins transects (Simmons et al., 2024).**

Intense geological investigations around the Timmins-Porcupine mine camp (Fig. 6) provides sufficient information of the greenstone assemblage thicknesses and deformed geometries (e.g., Ayer et al., 2005; Snyder et al., 2008) to permit approximate palinspastic restoration of the original sections (Fig. 7). Mapping demonstrates that the folds and thrusts in this

area are 3-dimensional, but most cumulative horizontal shortening was north-south oriented (Bleeker, 2015), thus broadly parallel to the orientation of the chosen cross section constrained by seismic reflectors. One permissible restoration (Fig. 7 top) indicates about 40 km of north-south shortening within the greenstones and metasediments during D2 and an early-D3 phase of south-verging, thin-skinned thrusting on a family of thrust faults related to the low-angle Pipestone fault (Bleeker, 2015; Haugaard et al., 2021; Adetunji et al., 2025). A lesser amount of horizontal shortening occurred immediately to the

south during later D3 north-verging, thick-skinned thrusting on thrusts related to the Porcupine-Destor fault (Figs. 7, 8) (Adetunji et al., 2025).

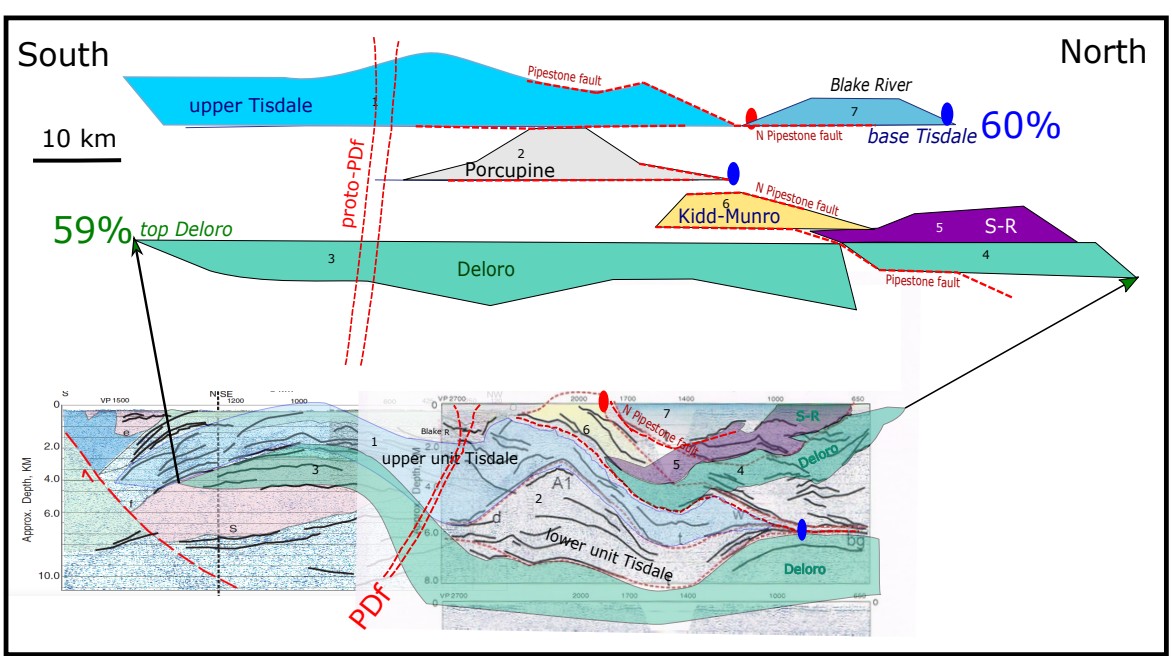

**Figure 7. Crawchest geologic cross section: palinspastically restored (a) and interpreted (b) based primarily on Snyder et al. (2008)**
**with their seismic sections shown. Corresponding assemblage names (e.g., Tisdale), numbered tectonic blocks (e.g., 2) and piercing points (upright blue ovals) indicated in both a and b. Estimated horizontal lengths at two assemblage contacts indicate final length has about 60% of the length of the original. PDf is the Porcupine–Destor fault zone.**

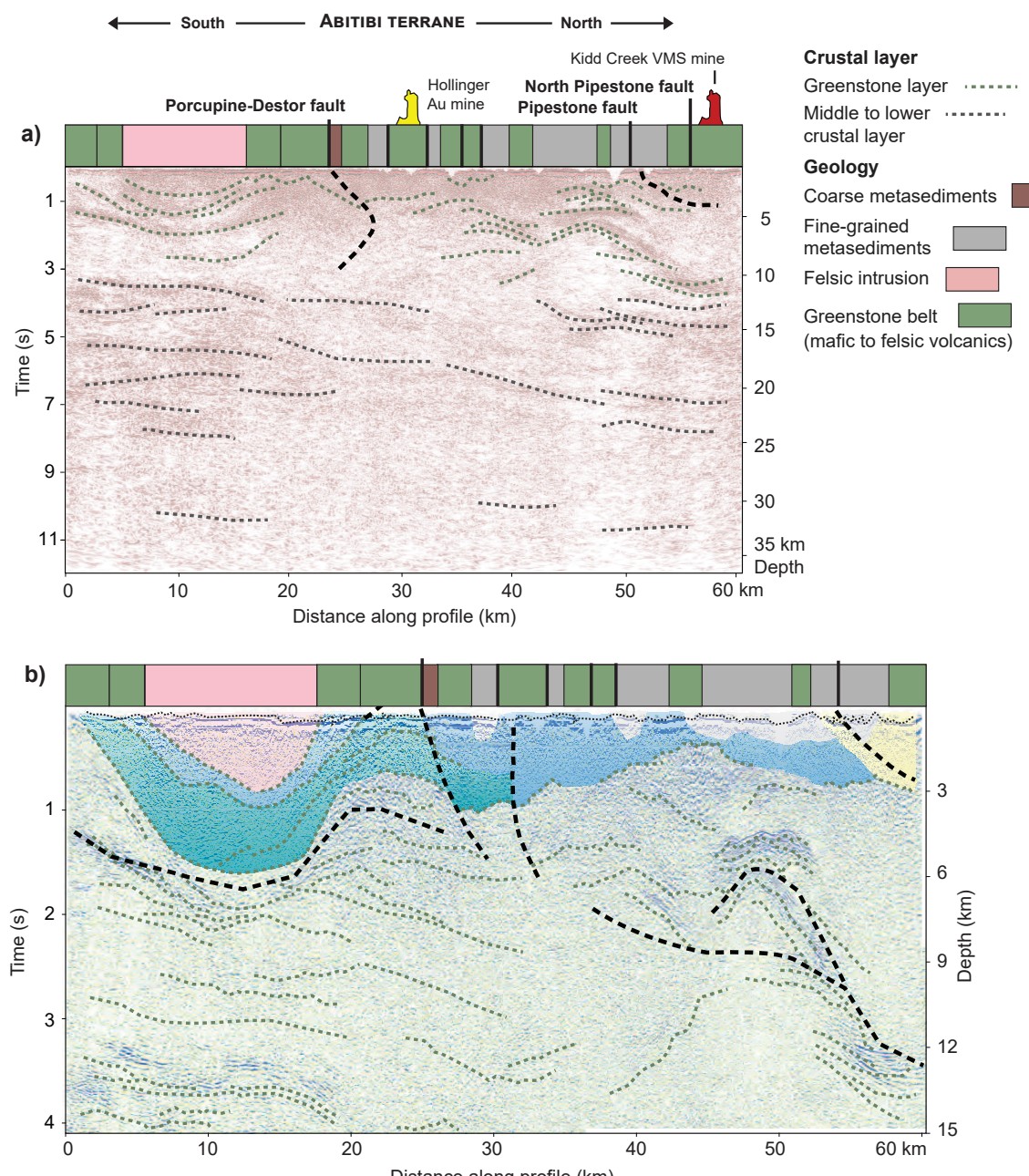

**Figure 8. Deep seismic reflection section across the Porcupine–Destor fault zone and Timmins gold camp of the Abitibi terrane: the Metal Earth South Timmins transect (Fig. 1). Line located in Fig. 6. (a) Strip geology and interpreted seismic reflection section (Simmons et al., 2024; Adetunji et al., 2025); and (b) detailed (note vertical exaggeration) interpreted seismic reflection section of upper crust (Snyder et al., 2008). This section illustrates strongly layered reflectors (dotted black form lines) throughout much of**
310 **the (middle) crustal layer beneath thrust and fold belt greenstones. This interpretation differs from Adetunji et al. (2025) because of the additional consideration of local 3-D structure in the near-surface gleaned from confidential mine observations.**

About 50 km to the east in the Matheson area (Fig. 1), drilling and seismic section interpretation indicate that, regionally, the Porcupine-Destor fault dips about 30-40° to the south whereas the south verging Pipestone thrust coincides at the surface with the north boundary of Porcupine assemblage and dips to the north (Snyder et al., 2008; Haugaard et al., 2021). These thrusts can be identified using seismic reflectors within the upper crustal 'greenstone' layer, but not so clearly within the mid-crust (Fig. 8). Steeply south-dipping zones of increased conductivity and disrupted or highly attenuated reflections at 15-30 km depths observed on several geophysical transects suggest continuation of the thick-skinned faults at depth.

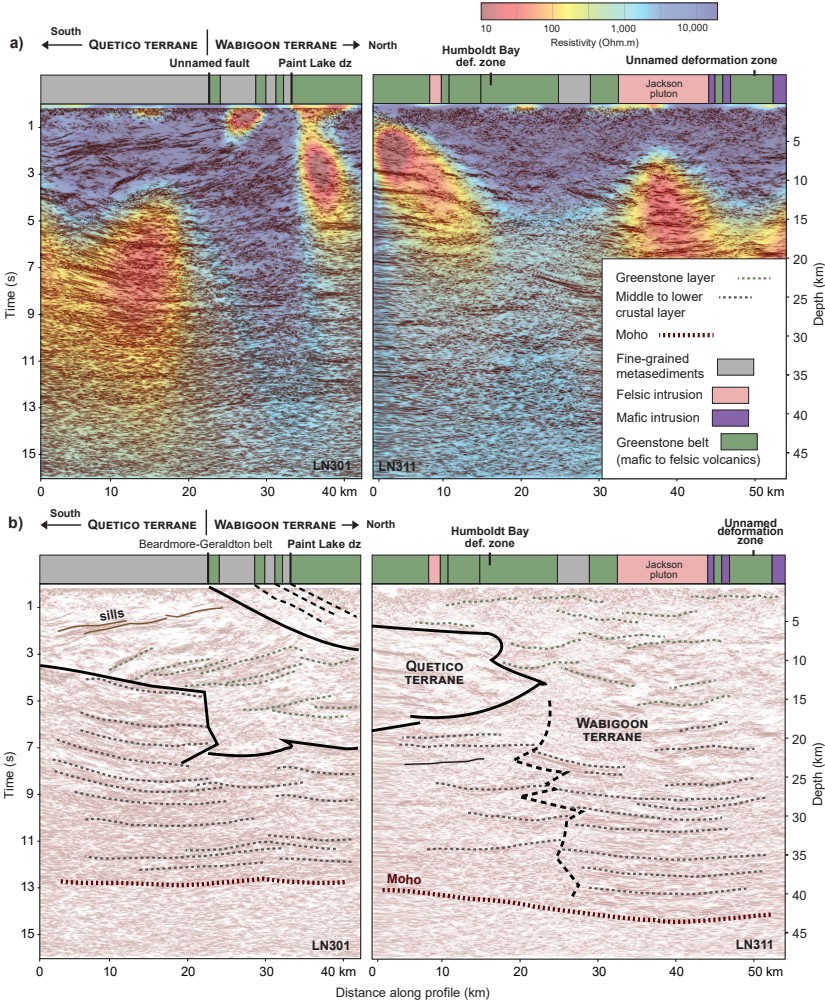

**Figure 9. Deep seismic reflection section across the Beardmore-Geraldton Belt: the southern part of the Metal Earth Geraldton transect (Fig. 1). (a) Strip geology, seismic reflection section, and conductivity model (Simmons et al., 2024); and (b) interpreted seismic reflection section. This section illustrates thrusting/wedging within the uppermost crust of the Wabigoon terrane southward over Quetico metasedimentary rocks and layered, gently north-dipping reflectors in the middle crustal layer of the Wawa terrane. Moho at 38-40 km is inferred from 3D Ps receiver function analysis (station GTO in Snyder et al., 2022). Greenstone assemblages range from 3.00 to 2.72 Ga.**

(Haugaard et al., 2021). Although crustal conductivity proximal to the Porcupine-Destor fault is three-dimensional and heterogeneous, the footwall generally appears more conductive in the upper 10 km and can help map the fault at depth (Fig. 10b) (Adetunji et al., 2025). Further east, near the Larder Lake transect, passive seismic studies identified a prominent seismic discontinuity dipping about 30° to the south immediately beneath the Moho (Fig. 10a). This Ps seismic discontinuity may represent a yet deeper continuation of the Porcupine-Destor fault that was displaced/offset from its shallower segments by shearing within the middle and lower crust (Naghizadeh et al., 2022).

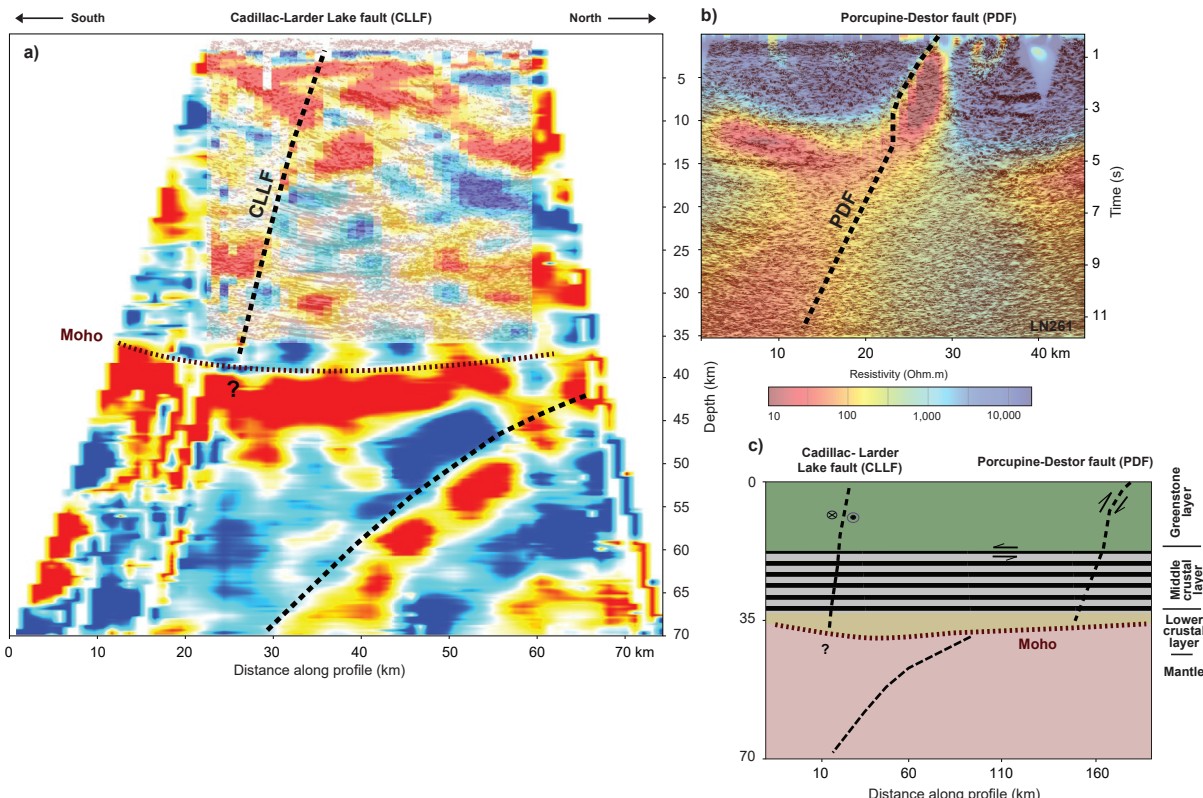

**Figure 10. (a) a P-S convertibility section for the Larder Lake transect from receiver function analysis of the passive seismic data (Naghizadeh et al., 2022). This image is an average of all the frequency bands. Hotter colors are positive convertibility amplitudes and cooler colors are negative. CLLF is the Cadillac-Larder Lake fault zone. (b) seismic reflection section and conductivity model of the Metal Earth Matheson transect (Simmons et al., 2024). (c) Structural interpretation of P-S section using several nearby seismic reflection sections for a more regional context. Sense of displacement shown is for peak D3 strain, all these faults experienced displacements with the opposite sense before or after D3.**

Further to the east, a seismic section north of the Malarctic gold mine and Cadillac-Larder Lake fault zone (CLLF) (Bedeaux et al., 2017) helps to project several surface faults at depth: the South Manneville, Aiquebelle, and Abcourt faults all appear as low-angle faults within the greenstone rocks of the Blake River assemblage (Fig. 5). The overall northward dip could indicate that these faults form a family of early D3 structures related to the Pipestone fault.

In the Wabigoon terrane a seismic section across the Beardmore-Geraldton Belt (Fig. 1) shows a trio of north-dipping, low-angle thrusts (Point Lake zone) where greenstones of the Wabigoon terrane are thrust southward and imbricated with more conductive metasedimentary rocks of the Quetico basin (center of Fig. 9) Tóth et al., 2025). These thrusts represent D3 north-south shortening observed throughout the southern Superior craton (considered D2 by Tóth et al., 2023). Its extent into the mid-crust suggests the belt represents a significant southern boundary of the Wabigoon terrane.

### 3.3 Terrane boundaries and Moho offsets

Surface faults that penetrate the lower crustal layer are rare (Stern and McBride, 1998). This layer is largely defined by its relationship to the Moho discontinuity. Due to their higher frequency content, reflection profile sections seldom unambiguously resolve the Moho discontinuity but do enable assessment of the structure of the lowermost crustal layer (Cook et al., 2010). Interpretations of several Lithoprobe seismic reflection sections within the Superior craton identified reflections dipping at low angle from the lowermost crust into the uppermost mantle, associated with a step in the inferred Moho depth (e.g., Calvert et al., 1995; White et al., 2003; van der Velden and Cook, 2005). In addition to two Lithoprobe refraction surveys (Grandjean et al., 1995; Musacchio et al., 2004), isolated teleseismic stations (Frederikson et al., 2007; Schaeffer and Lebedev, 2014) provide sparse multi-azimuthal estimates for Moho depth across the central Superior craton, ranging from 34 to 47 km depths (Snyder et al., 2021). Teleseismic stations near the Quetico/Porcupine basins have associated Moho depth estimates greater than 44 km; stations to the north and south generally have Moho depths of 39 to 42 km, values typical of Archean crust worldwide (Abbott and Mooney, 2013). The estimates greater than 44 km thus may represent thickened crust typically found along a linear tectonic feature such as a major terrane boundary.

Seismic reflection profiles in the southern Superior craton have also been interpreted to indicate similar local Moho topography of a few kilometers. Shallowly north-dipping reflectors in the middle and lower crust characterize these seismic sections that cross terrane boundaries between juvenile and older (with respect to 2.7 Ga) terranes. Such north-dipping reflectors occur near the Moho in the eastern Superior craton beneath the Abitibi-Opatica terrane boundary (Fig. 1) (Calvert et al., 1995; Matthieu et al., 2020; Daoudene et al., 2022). Near Chibougamau, the Barlow–North Bordure faults appear to form the terrane boundary (Fig. 3). Prominent southward dipping reflectors in the upper crust combined with northward dipping reflectors throughout the mid- and lower crust define an overall crustal wedge geometry (Matthieu et al., 2020). The 2.82–2.70 Ga Opatica crust wedges apart the 2.75–2.70 Ga Abitibi crust (Fig. 3b). The hanging wall block of the Barlow fault has much greater conductivity than its footwall (Fig. 3a).

Near Sturgeon Lake in the western Superior craton, similar reflectors occur beneath another terrane boundary (Fig. 2). These features have been interpreted as subduction, or at least underthrusting, of the southern 3.5–2.26 Ga Winnipeg River terrane beneath its northern neighbor, the 2.77–2.71 Ga Wabigoon terrane (White et al., 2003; Sanborn-Barrie and Skulski, 2006; Ma et al., 2021). Here an ambiguous Moho offset coincides with a marked change in seismic velocities within the lower crustal layer (Mussachio et al., 2004). The South Sturgeon and Sturgeon Lake faults, 200 km to the southeast, represent the

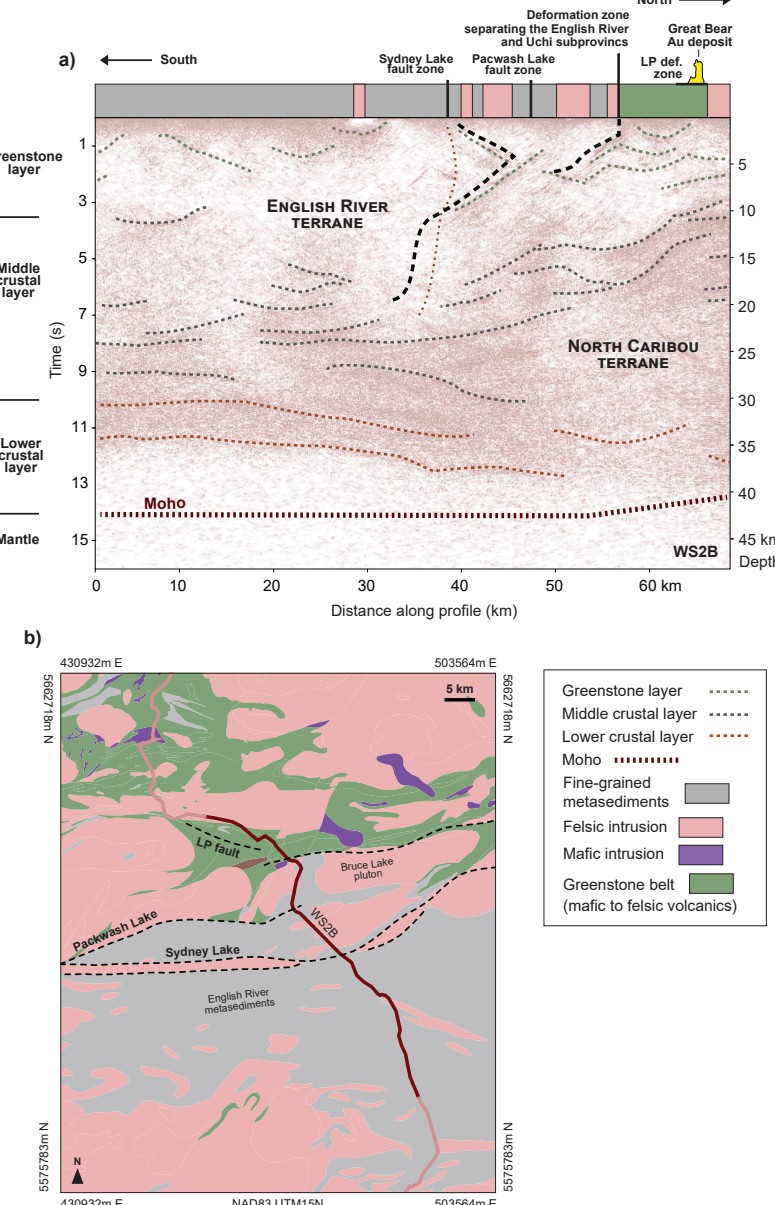

**Figure 11. Deep seismic reflection section across the terrane boundary between the North Caribou terrane and English River basin: the central part of the Lithoprobe Western Superior line 2B (Fig. 1). (a) re-interpreted seismic reflection section; (b) geological location map. Seismic section and Moho as described in Calvert et al. (2004). The Sydney Lake fault is interpreted to mark the terrane boundary. Moho location is taken from Musacchio et al. (2004).**

Upper crustal terrane boundary between the Winnipeg River and Wabigoon terranes (Percival et al., 2012). The Sturgeon Lake fault was interpreted previously by Ma et al. (2021) to cut vertically through the entire crust and to offset a horizontal reflector interpreted as the Moho. Our new alternative interpretation of a moderately northward dipping geometry is based on

the lack of steeply aligned truncations of reflectors anywhere within the zone between the Sturgeon Lake and South Sturgeon

faults (Fig. 2). In 3D, a large lower-crustal conductor parallels the Sturgeon Lake fault zone and projects into the upper crust where the two faults intersect.

About 115 km to the north, the Sydney Lake fault forms the terrane boundary between the Uchi subprovince of the North Caribou terrane and the Winnipeg River terrane, here covered by the English River basin (Fig. 1) (Hrabi and Cruden, 2006; Hynes and Song, 2006; Percival et al., 2012). Calvert et al., (2004) previously interpreted a nearly vertical fault geometry

within the upper crust that ends at a gently south-dipping horizontal reflector at 20 km depth (dashed orange line in Fig. 11). Our new interpretation of wedge geometries relies on the presence of angularly intersecting reflectors and a lack of steeply aligned reflector truncations beneath the surface fault zone.

The Quetico fault, 300 km to the southwest, parallels a terrane boundary between the Wabigoon and Quetico terranes between Lake Superior and Rainy River, Ontario (Figs. 1 and 12) (Percival et al., 2012; Hendrickson, 2016). The segment of

395 the Quetico fault near Atikokan, 250 km to the east of Rainy River, was interpreted previously as listric: vertical within the upper half of the crust and becoming a gently north-dipping reflector defining the lowermost crust (White et al., 2003). Our interpretation of moderate northward dipping fault geometries at Rainy River, consistent with the Beardmore-Geraldton Belt interpretation further east (Fig. 9), notes the lack of vertically aligned truncations of reflectors within the inferred greenstones and Porcupine-like (2.72 Ga Warclub Group) metasediments that underlie the Quetico fault zone here (Fig. 12).

Hendrickson (2016) modeled aeromagnetic data that he used to interpret a broad, steep-sided (~80°) synform structure to the Quetico basin within the uppermost 5 km of the crust. The inferred steep banded iron formation and mafic units may border the Quetico fault in the top few kilometers where incompletely migrated (in 3D) reflections lie, but near-horizontal reflectors at 4–10 km depths appear more reliable indicators of structure within the Quetico basin (Fig. 12).

### 3.4 Late or post-tectonic fault displacements

Late (2.665–2.595 Ga) movements on faults across the southern Superior craton vary locally from area to area but generally have strike-slip displacements that appear to accommodate continuing, lesser amounts of north-south shortening (Percival et al., 2012; Bleeker, 2015; Bedeaux et al., 2017; Dubé and Mercier-Langevin, 2020; Mercier-Langevin et al., 2020). Remobilized gold and sulfides, sanukitoid plutons, two-mica granites, and small syenite or pyroxenite bodies all indicate a significantly more hydrous geochemistry (Mole et al., 2021) and voluminous inferred fluid movement, perhaps through fault

zones (Hill et al., 2021, 2025; Roots et al., 2022). Few fault displacements of this type have been recognized uniquely on Metal Earth seismic reflection sections, probably because these (D4 and D5) displacements are relatively small compared to the main (D3) north-south shortening strains.

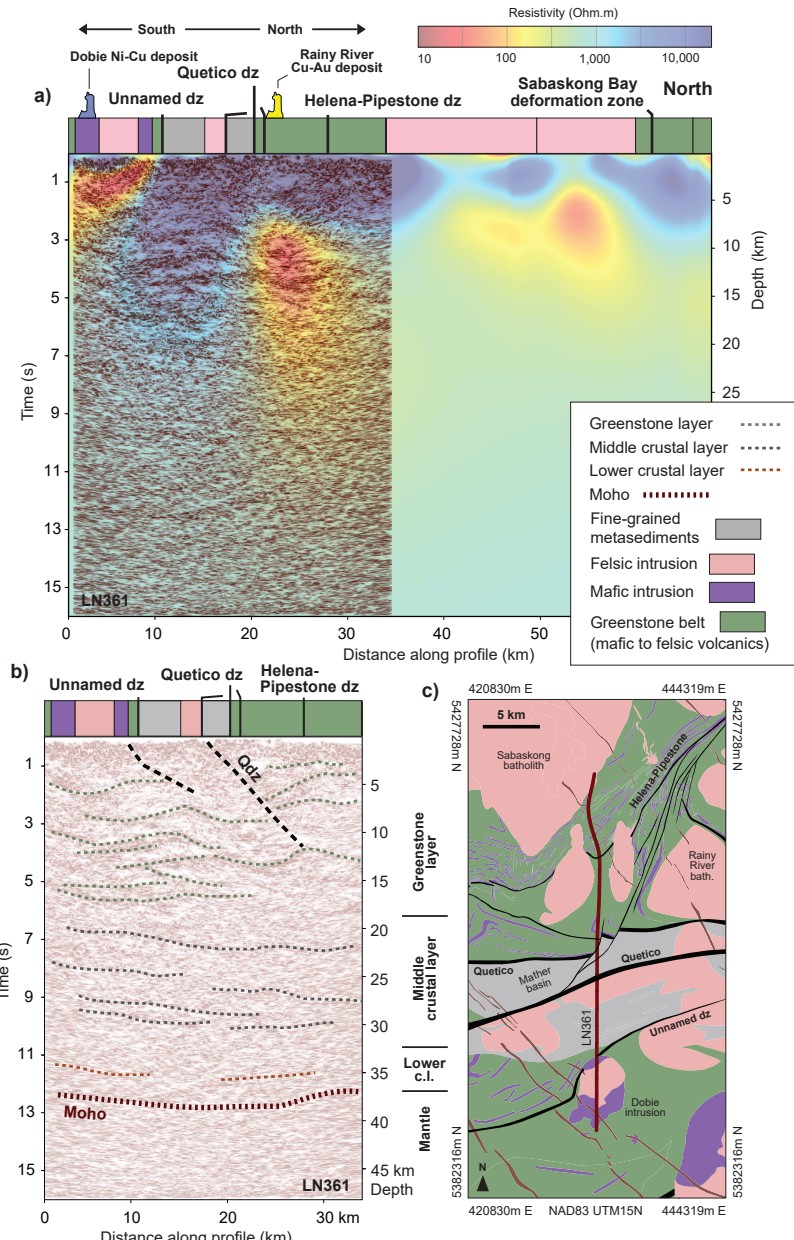

**Figure 12. Deep seismic reflection section across the Quetico fault zone within the Wabigoon terrane: the Metal Earth Rainy River transect (Fig. 1). (a) Strip geology, seismic reflection section and conductivity model section (Simmons et al., 2024); (b) interpreted seismic reflection section; (c) Location geological map with profile location (red line). This section illustrates broad folding and thrusting within the upper crust and layered reflectors in the mid- to lower crust. Moho at 37 km is inferred from regional seismic refraction models (Musacchio et al., 2004). Qdz is the Quetico deformation zone, the interpreted terrane boundary (White et al., 2003). Greenstone assemblages range from 2.78 to 2.70 Ga.**

## 4 Implications of these observations

Regional alignment of offsets and local topography on the Moho suggest that only two first-order lithospheric scale tectonic boundaries, perhaps sutures, occur within the southern Superior craton. Based on the location of Moho offsets, possible terrane sutures coincide with the southern boundary of the North Caribou terrane (Calvert et al., 2004) and the northern boundary of the Abitibi-Wawa terrane(s) (Calvert et al., 1995). In the southwestern Superior a series of micro-continental, primarily crustal terranes (Winnipeg River, Wabigoon, Marmion: Marmion superterrane of Bjorkman et al., 2024) lie between these main boundaries and above a SE-dipping mantle seismic discontinuity (Snyder and Thurston, 2024), perhaps similar to the present-day tectonic setting of the western Pacific (e.g., Macpherson and Hall, 2002) or the Mesozoic Pacific northwest of North America (Colpron et al., 2007; Cook et al. 2000). The southeastern Superior comprises the Opatica, Abitibi and Pontiac terranes and this simplification may indicate absence of microcontinents and one primary terrane boundary suture in the east (Daoudene et al., 2022). Next, we explore some fault specific implications of these large-scale tectonic trends.

### 4.1 Restoration of major fault geometries

Reconstructing relatively late deformation to remove associated strain and displacements makes it possible to recreate geometries of major faults before these displacements. Palinspastic restorations of the 2.72-2.66 Ga fold and thrust belt in the Timmins area (Dubé, and Mercier-Langevin, 2020) and estimated rotation on the Kapuskasing structure between 2.07 and 1.87 Ga (Evans and Halls, 2010) represent late deformations that affected the Superior craton (Fig. 13). Younger regional-scale deformation within the Superior craton, other than dyke intrusions, has not been recognized.

Geometrical patterns of Paleoproterozoic dyke swarms in the Superior craton and paleomagnetic studies of those dykes both indicate relative motion across the Kapuskasing Structural Zone (KSZ) that divides the craton into eastern and western parts (Fig. 1) (Evans and Halls, 2010). Euler parameters that optimally group the paleomagnetic remanence data from six dyke swarms with ages between 2.47 and 2.07 Ga indicate a restoration of the eastern Superior relative to a 'pinned' western part about an Euler pole at 51°N, 85°W, with a rotation angle of 14° CCW (Fig. 13). Dyke strikes on either side of the KSZ align best after application of 23° CCW distributed shear composed of 90 km of sinistral transpressional displacement. Half of this shear appears as distributed strain within the KSZ, half as oblique lateral thrusting (with NE-vergence) across the Ivanhoe Lake shear zone at ca. 1.90 Ga (Evans and Halls, 2010).

The D3 phase of north-south shortening at 2.72-2.66 Ga represents the vast majority of the crustal deformation recognizable today in the southern Superior (Dubé, and Mercier-Langevin, 2020). Structural measurements spanning the craton indicate a consistent D3 north-south horizontal shortening with systematic and slight deviations near major high-strain zones (e.g. Hrabi and Cruden, 2006; Bedeaux et al., 2017; Tóth et al., 2023, 2025). Benn (2006) argued that north-south shortening occurred via localized intense folding and flattening in the upper crustal mafic volcanic 'greenstones' and plutons, pure shear in the weaker mid-crustal 'TTG' gneisses, and large-scale imbrication in the stronger pyroxenite-peridotite rocks of the

lowermost crust and uppermost mantle (e.g. Fig.3). A detachment surface between basal 'greenstones' and basement gneisses would be needed to accommodate differential strain locally, and prominent basal reflectors at 10–25 km depths are often interpreted as this boundary zone (Fig. 7) (Benn and Peschler, 2005; Snyder al., 2008).

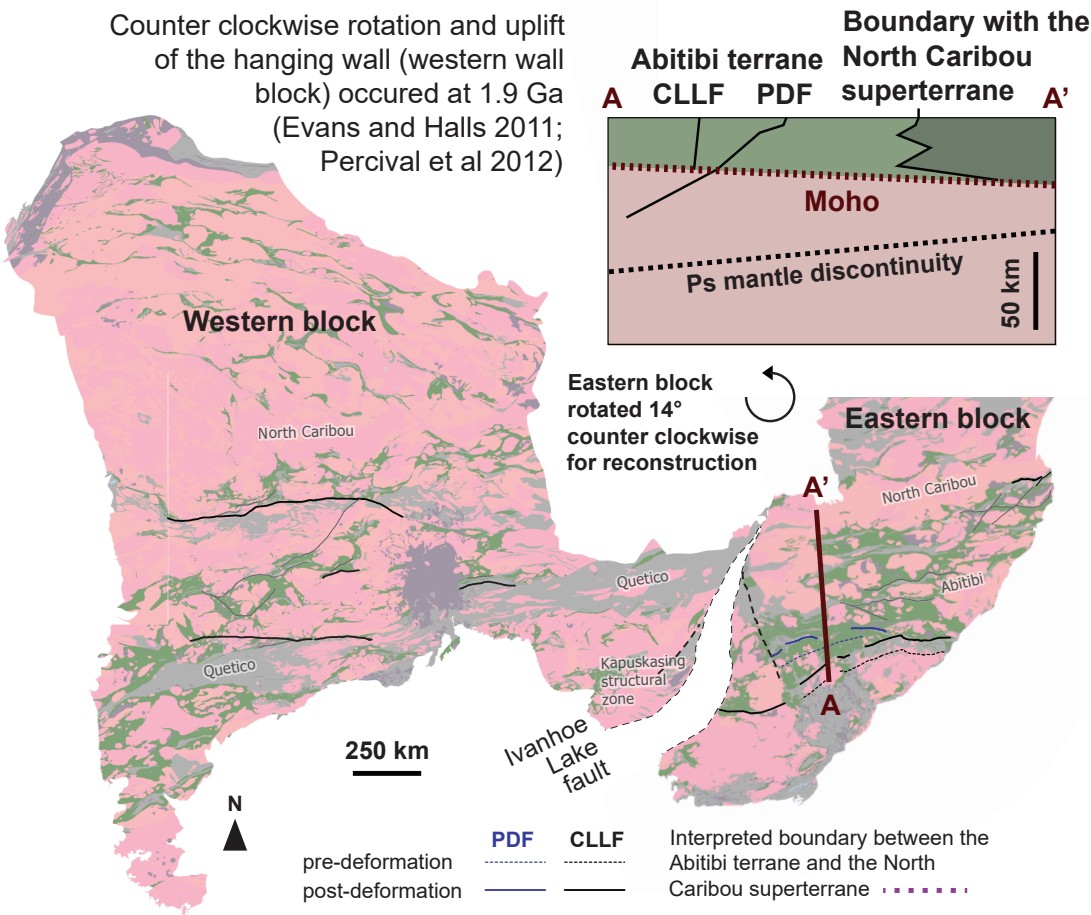

**Figure 13: Reconstructed fault geometries in map view, and in a north-south schematic cross section. The map shows a restoration of the eastern Superior relative to a 'pinned' western part about an Euler pole at 51°N, 85°W; with a rotation angle of 14° CCW. Distributed shear of 90 km of dextral transpressional displacement at ca. 1.90 Ga manifested half as distributed strain within the Ivanho Lake fault and half as oblique lateral thrusting (with NE-vergence) across the Ivanhoe Lake shear zone (Evans & Halls, 2010). The PDF and CLLF fault traces are drawn after rotation of the eastern block (solid lines) and again after additional restoration of N-S shortening as estimated in Fig. 6 (dotted lines). The Moho and Ps mantle discontinuity shown in the section were taken from Snyder et al. (2021). Geometrical patterns of Paleoproterozoic dyke swarms in the Superior craton and paleomagnetic studies of those dykes both indicate relative motion across the Kapuskasing Structural Zone (KSZ) that divides the craton into eastern and western parts (Fig. 1) (Evans and Halls, 2010). Euler parameters that optimally group the paleomagnetic remanence data from six dyke swarms with ages between 2.47 and 2.07 Ga indicate a restoration of the eastern Superior relative to a 'pinned' western part about an Euler pole at 51°N, 85°W, with a rotation angle of 14° CCW (Fig. 13). Dyke strikes on either side of the KSZ align best after application of 23° CCW distributed shear composed of 90 km of sinistral transpressional displacement. Half of this shear appears as distributed strain within the KSZ, half as oblique lateral thrusting (with NE-vergence) across the Ivanhoe Lake shear zone at ca. 1.90 Ga (Evans and Halls, 2010).**

Both recognized craton-scale deformations directly affect the two main shear zones internal to the Abitibi terrane: the Cadillac-Larder Lake (CLLF) and Porcupine- Destor (PDF) shear zones (Fig. 1). Restoring the rotation and horizontal shortening described above makes these faults more linear and parallel to each other in map view (Fig. 13). In cross section (Fig. 13), the south-dipping PDF and near-vertical CLLF geometries inferred from our seismic sections form paired, partitioned strain structures (Fig. 10c) typically recognized in convergent tectonic settings where the dipping fault absorbs boundary-normal shortening, the vertical fault oblique strains (e.g., Tikoff and Teyssier, 1994; McCaffrey, 2009; Haq and Davis, 2010), here at lithospheric scale. This lithospheric-scale strain partitioning may result from the possibly independent origin of these two faults. The pair were subsequently both active during peak north-south shortening.

We did not recognize similarly paired structures in other seismic sections discussed here, but crustal wedge geometries appear commonly at recognized major terrane boundaries (e.g., Figs. 3, 9 and 11). Implied underthrusting of only a few tens of kilometers of the Moho by mantle reflectors in these locations suggests that similar restorable horizontal shortening occurred within the overlying crust. The only other seismic observation constraints on thrust offset occur beneath the Winnipeg River terrane where Mussachio et al. (2004) and White et al. (2003) interpreted underthrust lower crust from refraction velocities and lower crustal reflectors, respectively.

## 4.2 Long-lived diverse history of some Superior faults

The southern Superior craton underwent a diverse history of deformation prior to its stabilization as a craton at about 2.65 Ga. Much of this strain history is recorded in its major fault zones, several of which will be highlighted here.

The Cadillac-Larder Lake fault (CLLF) forms a segmented, largely vertical east-west structure about 450 km long across the eastern half of the southern Superior craton (Fig. 1) (e.g., Bedeaux et al., 2017; Dubé and Mercier-Langevin, 2020). Several deep seismic reflection profiles that cross the fault zone indicate that it cuts deeply into the crust (Fig. 4). As discussed previously, the CLLF possibly first formed coeval with the upper crustal volcanic greenstones, TTG plutons and gneisses in a rift setting (Mole et al., 2021). Modern analogy with the Lau Basin of the western Pacific, specifically the Peggy's Ridge fault, suggests it formed a transform fault connecting triple junctions amid several micro-oceanic plates (Baxter et al., 2020). Field studies suggest that segments of the fault were active during all known deformations affecting the Abitibi terrane (Wilkinson et al., 1999; Bedeaux et al., 2017). Strong conductors at 60-90 km depths beneath the Kirkland Lake area and in several sub-vertical zones within the crust near this fault suggest that late migrating carbon- or sulfide-rich fluids left residuals (Wannamaker et al., 2014; Roots et al., 2022; 2024).

The Porcupine-Destor fault (PDF) parallels the Cadillac-Larder Lake fault zone about 200 km to its north (Fig. 1). Within the central third of its mapped length, seismic reflection sections constrain its upper crustal attitude as dipping 25-35° to the south within the greenstone layer (Snyder et al., 2008; Haugaard et al., 2022). Near Timmins the dip steepened during late differential horizontal shortening; in the mid-crustal (TTG) gneiss layer, conductors and attenuated reflectors suggest a

steeper southward dip (Fig. 8) (Haugaard et al., 2022; Adetunji et al., 2025). A Ps seismic discontinuity within the uppermost mantle near Kirkland Lake (Naghizadeh et al., 2022) may indicate the deeper continuation of this lithospheric-scale fault, now offset 70 km southward relative to its mapped surface location (Fig. 10). Considered together, the coeval deformation on the PDF and CLLF could represent strain partitioning during peak D3 N-S shortening, with the DPF absorbing the N-S shortening and the CLLF accommodating oblique components of convergence (e.g., Haq and Davis, 2010).

Dating deformation phases precisely remains inconclusive but permits the proposed consistent coeval strains. Within the central part of the CLLF, pluton emplacement, deformation, and coincident metamorphism occurred over a span of one million years (from 2.670 to 2.669 Ga) to over fourteen million years (from 2.675 to 2.661 Ga), during a regime of north-south, followed by northwest-southeast, regional shortening (Wilkinson et al., 1999). In the Timmins segment of the DPF, N-S shortening, thrust inversion of the extensional faults and deposition of upper Timiskaming sediments sometime near 2.670–2.669 Ga and a trachytic lava at $2.669.6 \pm 1.4$ Ga (Ayer et al. 2005).

The Sydney Lake fault in the western Superior craton (Fig. 1) was interpreted as nearly vertical within the upper half of the crust and its base truncated at a gently south-dipping horizontal reflector (Calvert et al., 2004). It probably represents a fault that similarly partitioned N-S shortening and oblique strike-slip strains between the North Caribou and Winnipeg River terranes during their protracted convergence between 2.75 and 2.65 Ga (Hrabi and Cruden, 2006; Hynes and Song, 2006). A new interpretation of reprocessed seismic reflectors indicates a series of interleaved thrust wedges of upper crust (Fig. 11). Two hundred kilometers to the south, the Quetico fault probably played a similar tectonic role near the Quetico and Marmion/Wabigoon terrane boundary (Fig. 12) (Hendrickson, 2016). Restoring the recognized multiple reversals of partitioned strike-slip along these upper crustal faults remains unresolved.

The Hawthorne Lake and Ivanho Lake faults comprise the main thrust faults bounding the Kapuskasing uplifted block and thus have major 2.07–1.87 Ga rotational displacements (Percival and West, 1994), but this Proterozoic strain may also mask earlier deformation, as yet unrecognized.

## 4.3 Fluid flow within fault zones

Conductivity models can aid fault interpretation of seismic reflectors by mapping subsequent (late) fluid flow pathways. Crustal or lithospheric-scale fault and shear zones appear to focus and concentrate large volumes of fluids that transport conductive minerals such as carbon and sulfides (Bleeker, 2015; Poulsen, 2017; Hill et al., 2021, 2025; Roots et al., 2022). In the Timmins area (Fig. 8), we interpret zones of reduced reflection amplitudes to originate from pervasive metasomatism associated with these fluids reducing density and wave-speed contrasts at rock layer contacts; enhanced conductivity resulted from residuals of carbon, metals, and sulphides left behind by the same metasomatic fluids (Snyder et al., 2008; Adetunji et al., 2025). These melts and fluids moved by slow (millions of years), pervasive (bulk) vertical percolation (Gibson and McKenzie, 2023), by steep, channelized flow across nearly horizontal rock layers (Snyder et al., 2008; Heinson et al. 2018)

or along contacts between moderately dipping rock layers, fractures or shear zones (Calvert and Ludden, 1995; Hill et al., 2021, 2025) as is more typically observed near Archean-Proterozoic boundaries (e.g., Aulbach et al., 2013).

Similar flow mechanisms transport melts upward from the mantle into the crust where some erupt directly to the surface as lamprophyres and kimberlites (e.g., Poulsen, 2017). Other fluids and melts stagnate in the mid-crust, possibly within enriched fluid reservoirs that have increased conductivity due to subsequent shearing along subhorizontal zones below the brittle-ductile transition (Hill et al., 2021). Flow to the surface then occurs either via high-strain zones characterized by near-vertical axial planes or in locations where intersecting major structures create low-strain regions where fluids can flow (Hill et al., 2025). The former structures are typical settings for small volume ore deposits whereas the latter host major deposits such as Olympic Dam, Timmins, and Malartic (e.g., Heinson et al., 2018; Dubé and Mercier-Langevin, 2020). Thus, specific structures and tectonic history at all levels of the lithosphere influence which parts of Archean cratons become relatively well-endowed with metals (and diamonds) and which do not. A period of peak metamorphic conditions and prominent horizontal shortening within the upper crust of the Superior craton at 2.75-2.68 Ga coincides with mobilization of some metals (Au and rare metals) and resetting of mantle ages (Smit et al., 2014). This suggests a mineralization system was active at this time throughout the upper lithosphere.

## 5 Conclusions

The geometry of major inactive and ancient faults at depth must be mapped using high-resolution geophysical surveys such as reflection profiling, supported by conductivity models based on magnetotelluric current observations. Here a subset of deep reflection profiles recently acquired in the Archean southern Superior craton of North America provides such data with which to map at depth some major shear zones, many of which host significant orogenic gold or VMS deposits. New (form) line drawings of prominent reflectors grouped these reflectors as upper crustal or mid-crustal based on their intersections: upper crustal reflectors intersected (crocodiles) whereas mid-crustal ones appear mostly parallel (lamellae). Brittle, mafic-felsic lavas ('greenstones') and plutons probably form the upper crust whereas ductilely deforming TTG gneisses form the middle crust.

Most faults are (re)interpreted as low-angle thrusts; a few notable exceptions appear as breaks in prominent reflectors aligned sub-vertically. Asymmetry of reflector density across major near-vertical shear zones such as the Cadillac-Larder Lake fault suggest a growth fault origin co-eval with eruption of the upper crustal greenstones; this fault zone possibly later reactivated as lithospheric-scale partitioned horizontal strain paired with the younger (D3) Porcupine-Destor fault. Low-angle thrusting probably occurred during the dominant phase of folding and horizontal shortening strain that occurred during craton-wide ($D_3$) crustal deformation, mineralization, and peak metamorphism at 2.72-2.66 Ga. Most deformation after $D_3$ was associated with modest lateral displacements not recognized on the seismic sections.

Conductivity models of the 3-D crust around the seismic profiles indicate that some faults acted as conduits for late metasomatic fluid flow, while others did not. This channelized flow possibly represents either a slow percolation process,

crossing individual reflectors within wide near-vertical zones of thrust wedging or utilized seismic pumping in relatively narrow faults. Proterozoic faults appear limited and local to the Kapuskasing uplift structure.

## Author contribution

DS: Writing – original draft preparation. JS, TJ: Figures**.** All: Writing – review & editing. DS, JA: Conceptualization and Funding acquisition. MN, SC: seismic data processing. AA, GH, ER: magnetotelluric data modelling and interpretation.

## Acknowledgements

Seismic reflection and magnetotelluric data can be obtained from the Metal Earth Hub website (https://metalearth.geohub.laurentian.ca/search?tags=Dataset); see also Simmons et al., 2024 for more information. A GOCAD 3D model of the Superior mantle is available as a NRCan Open File report 8756. Metal Earth is funded by the Canada First Research Excellence Fund. Metal Earth contribution MERC-ME-2025-23.

## Competing Interests

The contact author has declared that none of the authors has any competing interests.

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
