# Peer review of "Characterizing some major Archean faults at depth in the Superior craton, North America"

_EGUsphere, 2025_

## Author Response (AR1)

Response to Review —- 'Faults' paper: egusphere-2025-390

The paper by Snyder et al. attempts to provide an overview of several crustal faults crossed by various seismic and magnetotelluric surveys acquired in the Archean Superior craton as part of the Metal Earth project; much of the material is taken from previously published publications or the Metal Earth Atlas (Simmons et al., 2024). As presented, the paper comes across as a disjointed set of observations with quite poorly supported interpretations. My concerns fall into four categories: (a) fundamental misconceptions on the nature of Superior craton crust, (b) no clear well justified thesis to tie the interpretations together, (c) poor justification of the many individual seismic interpretations, and (d) a disorganised written presentation of the results with many paragraphs containing a jumble of disparate ideas.

>>We thank this reviewer for all the time and effort they put into this review. We understand most of the comments, but can address perhaps 3/4 of the very numerous concerns expressed due to lack of consistent, necessary observations across the Superior craton (e.g., an accepted framework for timing of the polyphase deformation affecting the entire south Superior craton). We note that the reviewer seems to have found nothing positive about the manuscript and has misread some clear statements. In all cases, we have tried to improve ambiguously worded sentences. Seismic/MT sections are taken from the Metal Atlas Atlas which itself has no geological interpretations; none of the interpretations shown in our manuscript have been previously published unless noted as such (e.g., Fig. 7b, 10a).

Superior crust: The authors assume that the crust is made up of three layers with an upper resistive (mafic) greenstone layer above a moderately conductive gneissic middle crust (lines 172-179); the greenstone layer is shown in the figures to be 12-18 km thick, based on the assumption that greenstones exhibit high electrical resistivity values. In fact on line 230, the upper crust is noted as extending to 25 km depth. Surely not in 40 km thick Archean crust! The fundamental flaw here is the association of high resistivity with the upper crust. This general assumption is incorrect, because granitoids can also exhibit high resistivities, as shown in the North Caribou terrane by Roots et al. (2024) and in the Pontiac by Roots et al. (2022). Figure 5 from the Malarctic transect also contradicts the three-layer model, showing a resistive upper crust where it is mostly granitic (La Motte and La Corne plutons). There is also absolutely no reason to believe that mafic greenstones are 18 km thick in Fig. 2 where the surface geology is shown to be mostly granitic, which is also indicated by seismic velocities of ~6.0 km/s in Fig. 2c. Furthermore, in Fig. 11 the greenstone layer is shown to include the English River metasedimentary belt, which exhibits amphibolite and granulite facies metamorphism. Though greenstones may be as thick as 10 km or so in the Abitibi belt where their surface expression is more extensive, the idea that the south Superior craton everywhere includes a >10km thick greenstone layer is fundamentally incorrect.

>>This concern is largely a mis-understanding of terminology, specifically our definition of the upper crust. The upper crustal layer is nowhere stated to consist of 100% mafic volcanic rocks; that it is not is clear from the most cursory look at surface geology which indeed has mafic volcanic greenstones, granitic plutons, and metasedimentary

rocks. Nor do we state that all high resistivity rocks are mafic greenstones; we do infer that the middle crust is less resistive. We have revised wording to indicate that an upper crust, characterized by its greenstones, exhibits high resistivities along much of the profiling, in contrast to a more uniformly, moderately conductive middle crust characterized by TTGs. Resistivities provide one guide to our crustal layers, but are not the sole defining characteristic; reflector geometries (intersecting versus parallel) are more important. We do not understand why mafic greenstone assemblages cannot total 18 or 25 km in thickness locally; we know of no observations that disprove that possibility. Fig.2c shows velocities of 6.5 km/s at 15 km depth, characteristic of a mix of felsic and mafic rocks.

Fault geometries: Although it's not well articulated in the confused text, it seems that a major point of the paper is to argue that sub-vertical crustal faults originated as (oceanic?) transform faults at the time the greenstones formed, and the low-angle thrusts are related to convergence during the various orogenic phases that affected the Southern Superior craton. No evidence is presented to support the former hypothesis other a vague allusion (lines 198-200) that subvertical faults in the Superior craton look like oceanic transform faults. Both oceanic crust and continental crust can support transform faults, e.g. the San Andreas fault in California, but that does not mean such continental faults have an oceanic origin. If the (unstated) motivation is that some key Superior craton faults originated in volcanic rocks in a subaqueous (oceanic?)setting, then it becomes important to distinguish between autochthonous and allochthonous greenstones; in the latter case, there is no reason why a fault would still be active and cut through the underlying continental crust after, say, obduction; in the former, why would a fault cut through underlying continental crust created and modified by multiple episodes of TTG magmatism? An argument supporting a long-lived fault history such as this needs evidence and justification, rather than a simple statement that some Archean faults look like those in a modern oceanic setting. In the case, of low-angle thrusts, it is likely that many are related to shortening across the Superior craton, but they can also be created in strike-slip regimes with varying stress regimes, so characterising the faults' vergence becomes important. If the bigger argument, is that the strike-slip and thrust faults are two elements of a strain-partitioning regime, then this needs to be clearly documented in terms of the fault geometry and timing of motion, which has not been presented.

>> Major conclusions concerning fault characterization have been emphasized in the abstract, introduction and conclusions. We are not yet willing to deduce that ALL sub-vertical faults are syn-volcanic. The reviewer is right to note our presumption that readers will know that it is now generally accepted that the Abitibi terrane formed in an oceanic setting (e.g., Mole, Dube refs); we have added qualifying sentences stating this more explicitly. A paragraph describing the characteristics of so-called 'growth' faults has been added to strengthen the hypothesis that the Cadillac-Larder Lake (CLL) fault originated as a leaky transform or early-stage rift at 2.69-2.75 Ga when volcanic rocks erupted and plutonic rocks were emplaced. We know of no evidence that individual mafic greenstone assemblages along the CLL were thrust significant distances.

Nowhere do we interpret the PDF as steeply dipping or necessarily syn-volcanic in origin; we show it as dipping at 30 degrees in the upper crust.

The San Andreas is a more complex example of strain partitioning and is only vertical to 12-15 km depth where it intersects a subducted oceanic slab (Furlong et al., 2024: https://doi.org/10.1029/2023TC007963). Our broad interpretation is indeed that the CLL fault later reactivated during strain partitioning; 2.675-2.669 Ga Timiskaming metasedimentary rocks found as panels in both the CLL and PD faults demonstrates that the faults were coeval during at least that 6 Ma period. That observation has now been added in a paragraph later in the text where partitioning is introduced and discussed.

Seismic interpretation: There are numerous problems with interpretation of the seismic reflection images, which are documented in more detail below. In general, seismic reflections do not stand out well from the background noise, which has been laterally smeared out into a subhorizontal fabric by the processing. Faults seems to be interpreted through laterally continuous reflections (Fig. 2a, 8a) and along the edge of migration artefacts (Fig. 4b), given a strange zig-zag geometry (Fig. 9b, 11a) intended to indicate wedging, and presented with two conflicting interpretations (Fig. 8a, 8b). In addition, the Moho is interpreted at unusually low depths (9.5 s or 30 km in Fig. 3b), at the bottom of the seismic data (Fig. 4b), and offset by 2 s (7 km) from the reflection Moho (Fig. 11a). None of this inspires much confidence in the seismic interpretation, or inferences there from.

>>We agree that the signal-to-noise ratio is less than in some deep seismic reflection displays with which this reviewer may be familiar, but we sought greater detail in our acquisition and processing. We invite comparison with early versions of some sections, published in Nazgadeh et al., 2000. We would then request criteria with which the reviewer quantitatively determined reflections were smeared horizontally; similarly with definition of migration artefacts. As per the "Smeared out" reflections in Metal Earth vs. Lithoprobe seismic data: ME data, having been acquired on Nodal receivers with no control on how much noise was getting recorded by nodes, led to generally noisier data in ME data compared to Lithoprobe data. In the 25-year gap between the Lithoprobe and ME acquisitions, we assert that the traffic on the roads and cultural noise probably increased. We would also note that Lithoprobe final sections have been dip-enhanced, often using aggressive dip coherency algorithms. The frequency content of Lithoprobe data is lower than ME data and that can generate the impression that ME data are smeared, but in reality it can be attributed to higher frequency content of the data. One could debate if the high frequency data are coherent enough in the Metal Earth data due to the increased noise level data. Nevertheless, our processing contractor did not try to remove the high frequency content and we kept them in the ME sections shown here. The high frequency part of the data can be filtered out and reduced to the resolution level of Lithoprobe data, but we feel that usable data and reflections exist in those high frequencies if proper zooming and investigation of target reflections can be carried out during interpretation. The reviewer would need to do this with the digital sections and point out exactly where our interpreted faults cross

coherent lateral reflectors (rather than just referencing an entire figure). Figures 8a & b are not conflicting if vertical scale change is considered; as now noted in caption. Our indications of Moho location are taken from independent (refraction, Ps) seismic analysis and not from the reflection sections; indeed, the discrepancies pointed out here are intended to demonstrate our lack of confidence in so-called 'reflection Moho' determinations that this seems to prefer. Sentences in the main text and figure captions were added to clarify this Moho definition.

The CLLF is presented as a convincing example of a subvertical fault extending to 30 km depth. However, in Fig. 4b, the fault does not continue upward to its surface location, because continuous shallow reflections are interpreted above the fault at 1-2.5 s, in contrast to the resistivity model that shows a steeply dipping conductor that is not discussed. I suspect that the truncation of reflections at >5 s is simply the edge of a migration artefact arising from amplitude variations in the unmigrated data. Note how deeper reflections exhibit greater lateral smearing. It's worth noting that Roots et al. (2022) also presented this seismic line together with Lithoprobe line 23 just to the south and showed apparently continuous subhorizontal reflections at this location below 4-5 s depth. Also why is the Moho interpreted at the bottom of the seismic data at 11.5 s? The seismic data are hardly convincing as they don't exist here! Apparently, the purpose of Fig. 8 here is to show the layered middle crust, but the figure also reveals two contradictory interpretations of the Porcupine-Destor fault. Is the PD fault meant to be a high-angle "synvolcanic" fault like CLLF? If so why is it in this section? Note that in Fig. 8a the PD fault cuts through a laterally continuous interpreted reflection at ~ 1 s, which seems contradictory.
In the P-S section in Fig. 10a, the interpreted CLLF cuts through a continuous body at ~10 km depth, and the interpretation seems problematic given that there are south-dipping artefacts subparallel to the CLLF in both crust and mantle. In Fig. 10b, the footwall to the PD fault is described as more conductive, which is clearly not the case below 15 km.

>>The reviewer is correct to be circumspect of dipping reflections in the uppermost seconds because muting of refractions and surface waves requires dense receiver-source arrays that were not used in the processing of this manuscript's figures. We do have alternative higher-resolution sections (published previously) in the CLLF case which guided our shallow interpretations. In general, fault geometries were based predominantly on reflectors at 2-6 s, not the near surface. See comments above concerning migration smearing, Lithoprobe resolution comparison and Moho definition. The sentence about PD fault conductivity was qualified in the text.

Written presentation: Though it may be challenging to provide an overview of the geology of the entire southern Superior craton, the section on the Geologic Setting makes no attempt to describe all the areas where the seismic lines are located. In paragraph 1, there is a summary of greenstone assemblages in the Abitibi belt, but no information is provided on greenstones in the western Superior. This paragraph, which presumably focusses on the granite-greenstone domains, also repeats seismic velocities described in the preceding paragraph, giving the impression of a confused

organisation of the paper. The next paragraphs on plutonic and gneissic domains are better written, but there is no clarification of the complexity found by Mole et al (2021), which perhaps provides an alternative to the successive accretion model(?).
Since the different stages of deformation are key to interpretation of the seismic images and the paper's conclusions, these stages need to be better described. Percival et al. (2006) noted 5 deformation stages, but are these the same phases across the entire south Superior craton, or do they just refer to individual orogens: North Superian, Uchian etc. If there is a model of successive southward accretion, how can D1 shortening affect both the North Caribou and Minnesota River Valley terranes that have not yet been accreted? Is there an accepted framework for timing of the polyphase deformation affecting the entire south Superior craton where the seismic lines are located? For example, is deposition of the English River sedimentary rocks in the west really considered coeval with the Porcupine basin in the east, as stated? Paragraph 1 of the "Faults observed…" section is also confused, as it starts discussing the seismic interpretation approach, but ends with very general statements about MT surveys. In fact, the resistivity models appear to have vey little influence on the interpretation of the seismic images, even though the resistivity models frequently appear clearer.  Shouldn't the paragraph in lines 162-166, be part of the motivation in the introduction rather than buried here? The paragraph at lines 167-179 is also confused with a description of how form lines are used in the interpretation, and then moves into characterizing the geology of the Kapuskasing Uplift.

>>The reviewer is perceptive in acknowledging the challenge in summarizing the geologic setting. Greenstone volcanic assemblages in the western Superior have only recently become rationalized with those in the Abitibi and qualifying sentences were added in captions. Deformation phases (D1-D5) have not been similarly rationalized to date (and is certainly not our purpose here), but the most important N-S shortening phase at 2.72 Ga has been observed across the southern Superior. The question of how many major sedimentary basins (Quetico, English River, Porcupine, Pontiac) are coeval is currently under debate, but indeed favoured by some Metal Earth geologists. Our intention was to simplify pan-Superior geology to what is widely agreed here in the Introduction, then discuss the many details and controversies of this intensely studied region where relevant to a particular fault or transect. This paper is intended for the proceedings of a seismic meeting, not as a geology synthesis.
We have rewritten parts of the Geological Setting and Faults Observed paragraphs with a view to ease this reviewer's confusion. We note that the seismic velocities are not repeated; the first set are indicated as global averages, the latter specific to the western Superior. Lines 162-166 were moved as suggested. The paragraph at lines 167-179 was intended as an introduction to the observation/interpretation methods used in subsequent discussions and thus repeats the key underlying three-layer assumption; therefore each element, although diverse at casual reading, contribute substantively to this more focused methodology description. This is largely a question of style.

Syn-volcanic faults section: Line 187-188 essentially stating that there are high-angle or low-angle faults within greenstones has no citation, but this is clearly a key

observation, if correct. (What is a fault-related structure?). Presumably, this comment is here because both fault types are syn-volcanic, but on lines 228-229 in the Syn-tectonic section low-angle faults are described as post-volcanic, which contradicts the earlier statement. Also why is there a discussion of the upper-middle crustal boundary here? Shouldn't this be in an earlier section on crustal layering together with some of the geological observations?

>>The initial statement is ours; can the reviewer suggest a relevant citation? No clearly syn-volcanic thrust faults were observed or presented in this manuscript, so we have led this reader astray in their presumption. Sentence reworded. Discussion of the upper-middle crustal boundary retained here because it is directly and specifically relevant to this interpretation and may not be a common pan-Superior feature.

Syn-tectonic faults section: Why is a description of the metasedimentary belts here and not in the Geological Setting section? On lines 216-217, can all these sedimentary units record D1 in a successive accretion regime? Deformation needs to be better described (see above). Extensional collapse of the metasedimentary belts is stated here to be at ~2.75 Ga, but Percival et al. (2012) in his Fig. 13 show the Winnipeg River and more southerly terranes belts to be apart at 2.72 Ga, with the Minnesota River Valley terrane apart at 2.69 Ga! So what's really going on?

>>Description of metasedimentary belts is enlarged here because of its specific relevance to the interpretations that follow. The assumption here (e.g., Snyder & Thurston, 2024) is that the metasedimentary belts possibly formed on a broad (N Caribou superterrane) continental shelf between accreting terranes, similar to the SW Pacific (S China basin, Philippines, Taiwan) today. These transgressive, diachronous basins potentially record D1 to at least D3. We know no evidence that requires wide inter-terrane oceans/seas isolating specific tectonic phases to single terranes.

Specific comments:
Fig. 5: There is a >5 km offset between the Moho on the reflection section and the velocity-depth profile. >>Yes. Believe the velocity profile. The velocity profile is from a survey north of the seismic section (clarification added to caption), so not necessarily inconsistent.

Fig. 6: Three seismic lines are shown on the geology map. Which one corresponds to Fig 8? >>Fig 8 and western line on Fig 6 clearly labelled as South Timmins line??

What is the purpose of introducing the palinspastic reconstruction in Fig. 7 and the associated geological discussion, which seems to come out of nowhere? This section just seems to be unnecessary and confusing. >>One of the stated main points of this manuscript is that one must restore later deformations if origins of faults are to be more reliably estimated. It seemed logical to introduce this after description of relevant seismic reflection sections. We are sorry this reviewer finds balanced cross sections useless and uninteresting; we feel not all readers will agree.

Line 290: You don't know these conductors are fractures. >>Deleted.

Line 290-291: What does it mean to say that these faults are akin to the PD fault, which has both senses of dip and is more steeply dipping? What is the purpose of this comment?>>The term 'family' implies similar structures, here thrusts. Where does the PD dip northward, except perhaps in uppermost km near Timmins? Intent is to interpret the South Manneville, Aiquebelle, and Abcourt faults as all splays off the PD fault—now clarified in text.

Line 292-296: Note the publication by White et al of seismic data from this area. The metasedimentary rocks are described as conductive, but Fig. 9 clearly shows rocks under the metasedimentary Quetico belt are resistive above 4 s. Where are the three low angle-faults? Is this something to do with the zig-zag feature? >>Yes, broken line complicates relationships. Three faults now redrawn and referenced explicitly as the Paint Lake zone shown in Fig 9. Metasedimentary rocks interpreted as underthrust (not at surface) are conductive within the Wabigoon terrane crust in the center of the figure —-clarified in text.

Line 299-300: This comment is not true and contradictory. The reflection Moho can commonly be inferred from good quality deep crustal reflection sections, though clearly it is a challenge with a lot of the Metal Earth seismic data! Note the clear reflection Moho on the Lithoprobe line in Fig. 11. >>Do disagree here. The Seismix meeting's community of seismologists has discussed this point for decades. Refraction survey Moho determinations are considered the gold standard as it is how this first-order seismic discontinuity was originally defined. Teleseismic Ps conversions (e.g., H-k method) use similar wavelengths. Reflection profiling 'sees' much finer structure and therefore changes in reflector density can be displaced from main velocity gradient/ step; although in many places they do coincide. Consideration of rock types, specifically garnet granulites that have mantle velocities, further complicates this issue. The intended point here, in agreement with Cook et al. (2010), is to use caution when interpreting a Moho.

Fig. 11. Why is the Moho interpreted at 14 s? Note the incorrect depth scale, which implies that the velocity in the upper crust is the same as the upper mantle, because the 5 km depth interval is the same when the time scale is linear. English River is a metasedimentary belt, not a greenstone layer, as implied by the left hand annotation. Another poorly justified zig-zag fault! >>Moho shown is taken from a nearly coincident refraction survey (Musacchio et al., 2004) per above reply. Depth scale is approximate as it is a constant velocity conversion, but the Moho is not greatly affected by this approximation because the major velocity change occurs below it; general comment on time-to-depth conversion added to Figure 2 caption. The English River terrane is clearly labelled as having near-surface fine-grained metasediments in both map and strip geology; neither the depth extent of the metasediments nor what rock type lies below are known. The reviewer is invited to visit this digital Atlas section online and zoom into the reflections.

Line 312: Abiitibi spelling >>corrected

Line 345: Where is the downward extension of the Sydney Lake fault? Does it follow the zig-zag trajectory? How is this justified as a fault? How is strike-slip motion of the Sidney Lake fault accommodated along one of these zig-zag structures? >>No (right or left) justified reflector terminations occur with which to interpret a steeply dipping fault beneath the surface trace of the Sydney Lake fault, instead our interpretation links nodes where reflectors intersect. Thrust faults commonly have steep lineations but also later sub-horizontal lineations that imply strike-slip displacements on the same surface.

Line 361-364: There is no need to introduce fluid movement here. It's just an unnecessary distraction. >>Intention was to strengthen late fault geometry interpretation by assuming conductors can represent former fluid pathways that left residuals of graphite or sulfides; such fluid flow is thought to have last occurred during peak metamorphism at 2.9 Ga. In places, such residuals occur in fractures, elsewhere in more diffuse sub-vertical flow channels.

Line 372-373: This is unnecessary as the modification is not described. >>Paragraph deleted.

Line 378: If these are all "crust-only" terranes, what is the nature of the underlying upper mantle? >>North Caribou lithosphere, as described in more detail in the reference.

Line 403: 10c not 10b >>Corrected.

Line 405: Suggesting the CLLF and PD are syn-volcanic faults controlling lithospheric strain is wild speculation with absolutely no justification in this paper, as far as I can see. >>The intended meaning was that both faults originated independently, only the CCLF probably early during mafic lava eruptions. Zones of relatively low strength; the faults then re-activated as a partitioning pair during peak N-S convergence. Rewritten to convey this intent more clearly.

Line 407: How was the amount of underthrusting inferred? >>Observed Moho offsets in reflection sections, refraction and gravity models, and teleseismic discontinuities required only a minimum of a few kilometers. Greater amounts of underthrusting by uppermost mantle lithosphere remains possible, as previously interpreted as subduction by some workers, but not directly observed and thus not required.

Line 409-410: What does this mean? >>Oceanic lithosphere that formerly separated the accreted terranes had to go somewhere during terrane convergence. The width of these oceans is unknown because these (presumed) underthrust slabs have not yet been consistently observed; but see White et al. and Musacchio et al., 2004 for possible evidence in the form of interpreted pyroxenite slabs. Text was greatly expanded to address this unclear statement.

Line 432: I have seen no convincing seismic evidence that the CLLF cuts through the entire crust. See Roots et al. (2022). There has been no real justification presented for the initial formation of the CLLF as synvolcanic. >>Sorry that Fig 4 is unconvincing; is the concern primarily with the uppermost and lowermost few kilometers of crust? Reworded to say 'cuts deep into the crust'. Roots et al. (2022) used lower frequency data. The repeated syn-volcanic origin concern was addressed above in the relevant section.

Line 460-479: This section appears to largely be a reproduction of previous MT results. >>Not sure what is exactly the concern here. Plagarism? Discussions often summarize relevant previous studies.

Line 491: Kenoran orogeny is mentioned twice, once in the abstract and once in the Conclusions, but the term is never defined. >>It is a more meaningful name to some geologists, but just an alternative to the D3 label. Now also mentioned in the Geological Setting section.

---

## Referee Report (RR1)

This paper presents a summary of the seismic and other geophysical studies of the Superior Craton. It reflects a compilation of Lithoprobe data and new data from the Metal Earth project.

It has received contrasting reviews, and I would say that my review sits between the two.

It is good to see the extent of work that has been done. However, this can only ever be an interpretation as the exact geometry of the geology in cratonic areas at depth is poorly constrained, so one would expect a degree of variation in the interpretations. The temptation is to overinterpret such results – for example the strong assertation that the vertical faults are leaky transforms – why not simply vertical accretion boundaries of more mafic material with more granitic material?

This result is an interpretation as there is no plate tectonic context, there is no strong geological context as in these Archean cratons can only be interpreted based on what one thinks might have been happening. The implication here is that there has been tectonic accretion and that the crustal structures represent that tectonic accretion process with imbrication of the relatively low angle thrusts.

Only where there is exposure of deep crust can some sort of constraints be placed and these can be inferred from the Kapuskasing section. It would have been useful to see some more direct comparisons with mid-crustal reflections from this section. I believe that a lot of the low angle reflectors are mafic sections like the amphibolites in the Kapuskasing structure. The same goes for some of the layered lower crustal sections that one sees outcropping from the Superior province in for example the Ungava region underthrusting the Proterozoic sections (see work from Lucas, St Onge etc..) These are very instructive on what one might interpret in the Superior geophysical section and how layering might develop during the accretion process.

The section on fluid flow in the crust is poor and comes through as an afterthought. The Metal Earth project was conceive=d to look at fluids and mineralisation in the cratons and their margins – an entire paper could be written about how the fluid pathways are preserved – especially in the southern Superior where there are significant greenstone belts, superimposed by Huronian rifting and fluids and even younger kimberlites associated with Phanerozoic rifting.

In general, given that the authors have replied to the critical comments the I would think the paper is acceptable for publication with some modifications. It would be worth seeing some less definitive assertions and a more nuanced interpretation that recognizes that we do not really have much idea of the exact processes of Archaean tectonic accretion.

---

## Author Response (AR2)

Reviewer 1

The manuscript compiles several seismic and magnetotelluric profiles in the Upper Archean Craton, revealing the geometry of major ancient faults, reconstructing their activity and reactivation over time, and their role in the craton's evolution.

The new version of the article is more clearly written and well structured, and leads the reader through the study with greater ease. The data presented, particularly the figures, strongly support the authors' interpretations and conclusions.

I have only two small points to make about the manuscript:

1. "TL layer" is mentioned on line 40, but its meaning is not stated until lines 109-110. Mention its meaning on line 40. >>TTG? layer; meanings added

2. On line 447, reference is made to figure 13b, but figure 13b does not exist. Correct to "fig. 13". >>Done

Reviewer 3

This paper presents a summary of the seismic and other geophysical studies of the Superior Craton. It reflects a compilation of Lithoprobe data and new data from the Metal Earth project.

It has received contrasting reviews, and I would say that my review sits between the two.

It is good to see the extent of work that has been done. However, this can only ever be an interpretation as the exact geometry of the geology in cratonic areas at depth is poorly constrained, so one would expect a degree of variation in the interpretations.

The temptation is to overinterpret such results – for example the strong assertation that the vertical faults are leaky transforms – why not simply vertical accretion boundaries of more mafic material with more granitic material?

>>Many of these faults have been (over)interpreted previously, some numerous times, so we tried to emphasize how our new interpretation differed and the broader tectonic implications. That perhaps sounded assertive. We have added qualifying phrases throughout and additional supporting observations for syn-volcanic transform faults.

This result is an interpretation as there is no plate tectonic context, there is no strong geological context as in these Archean cratons can only be

interpreted based on what one thinks might have been happening. The implication here is that there has been tectonic accretion and that the crustal structures represent that tectonic accretion process with imbrication of the relatively low angle thrusts.

>>We feel this reviewer is a tad too pessimistic. Although interpretation of seismic sections in neotectonic settings are greatly aided by earthquakes that define the fault zone and provide sense of offset, sections in all other geological settings can only provide deep interpretations without direct validation (rare drill holes excepted). Interpretation of Archean structures are indeed uniquely hindered by the lack of a widely agreed 'plate tectonic' paradigm for context.

Only where there is exposure of deep crust can some sort of constraints be placed and these can be inferred from the Kapuskasing section. It would have been useful to see some more direct comparisons with mid-crustal reflections from this section. I believe that a lot of the low angle reflectors are mafic sections like the amphibolites in the Kapuskasing structure. The same goes for some of the layered lower crustal sections that one sees outcropping from the Superior province in for example the Ungava region underthrusting the Proterozoic sections (see work from Lucas, St Onge etc..) These are very instructive on what one might interpret in the Superior geophysical section and how layering might develop during the accretion process.

>>The Percival & West 1994 synthesis volume provides arguably the best interpretable seismic sections from the Kapaskasing structure and show a series of linked low-angle thrusts that form crustal wedges. This albeit Proterozoic structure is one of the examples that initially inspired many of our interpretations of thrust-bound wedges in this manuscript. Their interpretation of a middle crust characterized by felsic intrusions into (mafic) amphibolites and localized horizontal shear zones we feel is consistent with our interpretations, although not directly relevant to faults. Our interpretation of thin-skinned folding in the Timmins area (Fig. 8) would be consistent with the described younger Ungava structures.

The section on fluid flow in the crust is poor and comes through as an afterthought. The Metal Earth project was conceived to look at fluids and mineralisation in the cratons and their margins – an entire paper could be written about how the fluid pathways are preserved – especially in the southern Superior where there are significant greenstone belts, superimposed by Huronian rifting and fluids and even younger kimberlites associated with Phanerozoic rifting.

>>All true and several papers by Graham Hill, referenced here, described this fluid flow process via preserved conductor geometries. This section was

included because many readers will wonder how the conductors shown on the Metal Earth Atlas sections in our figures relate to the proposed fault zone geometries. We interpret this fluid flow as not occurring exclusively in fault planes , but also by percolation through wider structural zones of disruption. Although probably thus an afterthought, we feel this is an important clarification.

In general, given that the authors have replied to the critical comments I would think the paper is acceptable for publication with some modifications. It would be worth seeing some less definitive assertions and a more nuanced interpretation that recognises that we do not really have much idea of the exact processes of Archaean tectonic accretion.
>>Nuance added in several places. A wise colleague once told me that interpretations (of a single seismic section) need only be permitted by other currently available observations/knowledge; they do not need to be either definitive or encompassing all possibilities.